# DiGress: Discrete Denoising diffusion for graph generation

**Clément Vignac**[*]
LTS4, EPFL
Lausanne, Switzerland

**Igor Krawczuk**[*]
LIONS, EPFL
Lausanne, Switzerland

**Antoine Siraudin**
LTS4, EPFL
Lausanne, Switzerland

**Bohan Wang**
LTS4, EPFL
Lausanne, Switzerland

**Volkan Cevher**
LIONS, EPFL
Lausanne, Switzerland

**Pascal Frossard**
LTS4, EPFL
Lausanne, Switzerland

## Abstract

This work introduces DiGress, a discrete denoising diffusion model for generating graphs with categorical node and edge attributes. Our model utilizes a discrete diffusion process that progressively edits graphs with noise, through the process of adding or removing edges and changing the categories. A graph transformer network is trained to revert this process, simplifying the problem of distribution learning over graphs into a sequence of node and edge classification tasks. We further improve sample quality by introducing a Markovian noise model that preserves the marginal distribution of node and edge types during diffusion, and by incorporating auxiliary graph-theoretic features. A procedure for conditioning the generation on graph-level features is also proposed. DiGress achieves state-of-the-art performance on molecular and non-molecular datasets, with up to 3x validity improvement on a planar graph dataset. It is also the first model to scale to the large GuacaMol dataset containing 1.3M drug-like molecules without the use of molecule-specific representations.

## 1 Introduction

Denoising diffusion models (Sohl-Dickstein et al., 2015; Ho et al., 2020) form a powerful class of generative models. At a high-level, these models are trained to denoise diffusion trajectories, and produce new samples by sampling noise and recursively denoising it. Diffusion models have been used successfully in a variety of settings, outperforming all other methods on image and video (Dhariwal & Nichol, 2021; Ho et al., 2022). These successes raise hope for building powerful models for graph generation, a task with diverse applications such as molecule design (Liu et al., 2018), traffic modeling (Yu & Gu, 2019), and code completion (Brockschmidt et al., 2019). However, generating graphs remains challenging due to their unordered nature and sparsity properties.

Previous diffusion models for graphs proposed to embed the graphs in a continuous space and add Gaussian noise to the node features and graph adjacency matrix (Niu et al., 2020; Jo et al., 2022). This however destroys the graph's sparsity and creates complete noisy graphs for which structural information (such as connectivity or cycle counts) is not defined. As a result, continuous diffusion can make it difficult for the denoising network to capture the structural properties of the data.

In this work, we propose DiGress, a *discrete* denoising diffusion model for generating graphs with categorical node and edge attributes. Our noise model is a Markov process consisting of successive graphs edits (edge addition or deletion, node or edge category edit) that can occur independently on each node or edge. To invert this diffusion process, we train a graph transformer network to predict the clean graph from a noisy input. The resulting architecture is permutation equivariant and admits an evidence lower bound for likelihood estimation.

We then propose several algorithmic enhancements to DiGress, including utilizing a noise model that preserves the marginal distribution of node and edge types during diffusion, introducing a novel

---

[*]Equal contribution. Contact: `first_name.last_name@epfl.ch`

guidance procedure for conditioning graph generation on graph-level properties, and augmenting the input of our denoising network with auxiliary structural and spectral features. These features, derived from the noisy graph, aid in overcoming the limited representation power of graph neural networks (Xu et al., 2019). Their use is made possible by the discrete nature of our noise model, which, in contrast to Gaussian-based models, preserves sparsity in the noisy graphs. These improvements enhance the performance of DiGress on a wide range of graph generation tasks.

Our experiments demonstrate that DiGress achieve state-of-the-art performance, generating a high rate of realistic graphs while maintaining high degree of diversity and novelty. On the large MOSES and GuacaMol molecular datasets, which were previously too large for one-shot models, it notably matches the performance of autoregressive models trained using expert knowledge.

## 2 DIFFUSION MODELS

In this section, we introduce the key concepts of denoising diffusion models that are agnostic to the data modality. These models consist of two main components: a noise model and a denoising neural network. The noise model $q$ progressively corrupts a data point $x$ to create a sequence of increasingly noisy data points $(z^1, \ldots, z^T)$. It has a Markovian structure, where $q(z^1, \ldots, z^T|x) = q(z^1|x) \prod_{t=2}^{T} q(z^t|z^{t-1})$. The denoising network $\phi_\theta$ is trained to invert this process by predicting $z^{t-1}$ from $z^t$. To generate new samples, noise is sampled from a prior distribution and then inverted by iterative application of the denoising network.

While early models would directly predict $z^{t-1}$ from $z^t$ (Sohl-Dickstein et al., 2015), these models were difficult to train due to the dependence of $z^{t-1}$ on the sampled diffusion trajectories. Ho et al. (2020) considerably improved performance by establishing a connection with score-based models (Song & Ermon, 2019). They showed that when $\int q(z^{t-1}|z^t, x)dp_\theta(x)$ is tractable, $x$ can be used as the target of the denoising network, which removes an important source of label noise.

For a diffusion model to be efficient, three properties are required:

1. The distribution $q(z^t|x)$ should have a closed-form formula, to allow for parallel training on different time steps.
2. The posterior $p_\theta(z^{t-1}|z^t) = \int q(z^{t-1}|z^t, x)dp_\theta(x)$ should have a closed-form expression, so that $x$ can be used as the target of the neural network.
3. The limit distribution $q_\infty = \lim_{T \to \infty} q(z^T|x)$ should not depend on $x$, so that we can use it as a prior distribution for inference.

These properties are all satisfied when the noise is Gaussian. When the task requires to model categorical data, Gaussian noise can still be used by embedding the data in a continuous space with a one-hot encoding of the categories (Niu et al., 2020; Jo et al., 2022). We develop in Appendix A a graph generation model based on this principle, and use it for ablation studies. However, Gaussian noise is a poor noise model for graphs as it destroys sparsity as well as graph theoretic notions such as connectivity. Discrete diffusion therefore seems more appropriate to graph generation tasks.

Recent works have considered the discrete diffusion problem for text, image and audio data (Hoogeboom et al., 2021; Johnson et al., 2021; Yang et al., 2022). We follow here the setting proposed by Austin et al. (2021). It considers a data point $x$ that belongs to one of $d$ classes and $\boldsymbol{x} \in \mathbb{R}^d$ its one-hot encoding. The noise is now represented by transition matrices $(\boldsymbol{Q}^1, ..., \boldsymbol{Q}^T)$ such that $[\boldsymbol{Q}^t]_{ij}$ represents the probability of jumping from state $i$ to state $j$: $q(z^t|z^{t-1}) = \boldsymbol{z}^{t-1}\boldsymbol{Q}^t$.

As the process is Markovian, the transition matrix from $\boldsymbol{x}$ to $\boldsymbol{z}^t$ reads $\bar{\boldsymbol{Q}}^t = \boldsymbol{Q}^1\boldsymbol{Q}^2...\boldsymbol{Q}^t$. As long as $\bar{\boldsymbol{Q}}^t$ is precomputed or has a closed-form expression, the noisy states $\boldsymbol{z}^t$ can be built from $\boldsymbol{x}$ using $q(z^t|x) = \boldsymbol{x}\bar{\boldsymbol{Q}}^t$ without having to apply noise recursively (Property 1). The posterior distribution $q(z_{t-1}|z_t, x)$ can also be computed in closed-form using Bayes rule (Property 2):

$$q(z^{t-1}|z^t, x) \propto \boldsymbol{z}^t \, (\boldsymbol{Q}^t)' \odot \boldsymbol{x} \, \bar{\boldsymbol{Q}}^{t-1} \tag{1}$$

where $\odot$ denotes a pointwise product and $\boldsymbol{Q}'$ is the transpose of $\boldsymbol{Q}$ (derivation in Appendix D). Finally, the limit distribution of the noise model depends on the transition model. The simplest and most common one is a uniform transition (Hoogeboom et al., 2021; Austin et al., 2021; Yang et al., 2022) parametrized by $\boldsymbol{Q}^t = \alpha^t \boldsymbol{I} + (1 - \alpha^t)\mathbf{1}_d\mathbf{1}_d'/d$ with $\alpha^t$ transitioning from 1 to 0. When $\lim_{t \to \infty} \alpha^t = 0$, $q(z^t|x)$ converges to a uniform distribution independently of $x$ (Property 3).

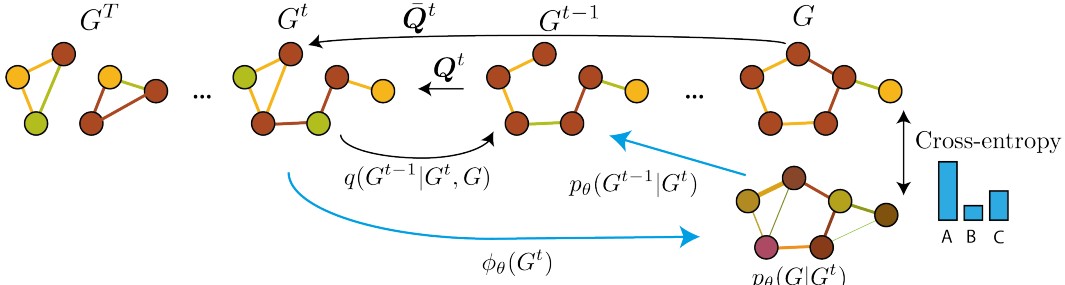

Figure 1: Overview of DiGress. The noise model is defined by Markov transition matrices $\boldsymbol{Q}^t$ whose cumulative product is $\bar{\boldsymbol{Q}}^t$. The denoising network $\phi_\theta$ learns to predict the clean graph from $G^t$. During inference, the predicted distribution is combined with $q(G^{t-1}|G, G^t)$ in order to compute $p_\theta(G^{t-1}|G^t)$ and sample a discrete $G^{t-1}$ from this product of categorical distributions.

The above framework satisfies all three properties in a setting that is inherently discrete. However, while it has been applied successfully to several data modalities, graphs have unique challenges that need to be considered: they have varying sizes, permutation equivariance properties, and to this date no known tractable universal approximator. In the next sections, we therefore propose a new discrete diffusion model that addresses the specific challenges of graph generation.

## 3 DISCRETE DENOISING DIFFUSION FOR GRAPH GENERATION (DIGRESS)

In this section, we present the Discrete Graph Denoising Diffusion model (DiGress) for graph generation. Our model handles graphs with categorical node and edge attributes, represented by the spaces $\mathcal{X}$ and $\mathcal{E}$, respectively, with cardinalities $a$ and $b$. We use $x_i$ to denote the attribute of node $i$ and $\boldsymbol{x}_i \in \mathbb{R}^a$ to denote its one-hot encoding. These encodings are organised in a matrix $\boldsymbol{X} \in \mathbb{R}^{n \times a}$ where $n$ is the number of nodes. Similarly, a tensor $\mathbf{E} \in \mathbb{R}^{n \times n \times b}$ groups the one-hot encoding $\boldsymbol{e}_{ij}$ of each edge, treating the absence of edge as a particular edge type. We use $\boldsymbol{A}'$ to denote the matrix transpose of $\boldsymbol{A}$, while $\boldsymbol{A}^T$ is the value of $\boldsymbol{A}$ at time $T$.

### 3.1 DIFFUSION PROCESS AND INVERSE DENOISING ITERATIONS

Similarly to diffusion models for images, which apply noise independently on each pixel, we diffuse separately on each node and edge feature. As a result, the state-space that we consider is not that of graphs (which would be too large to build a transition matrix), but only the node types $\mathcal{X}$ and edge types $\mathcal{E}$. For any node (resp. edge), the transition probabilities are defined by the matrices $[\boldsymbol{Q}_X^t]_{ij} = q(x^t = j | x^{t-1} = i)$ and $[\boldsymbol{Q}_E^t]_{ij} = q(e^t = j | e^{t-1} = i)$. Adding noise to form $G^t = (\boldsymbol{X}^t, \mathbf{E}^t)$ simply means sampling each node and edge type from a categorical distribution defined by:

$$q(G^t|G^{t-1}) = (\boldsymbol{X}^{t-1}\boldsymbol{Q}_X^t, \mathbf{E}^{t-1}\boldsymbol{Q}_E^t) \quad \text{and} \quad q(G^t|G) = (\boldsymbol{X}\bar{\boldsymbol{Q}}_X^t, \mathbf{E}\bar{\boldsymbol{Q}}_E^t) \tag{2}$$

for $\bar{\boldsymbol{Q}}_X^t = \boldsymbol{Q}_X^1...\boldsymbol{Q}_X^t$ and $\bar{\boldsymbol{Q}}_E^t = \boldsymbol{Q}_E^1...\boldsymbol{Q}_E^t$. When considering undirected graphs, we apply noise only to the upper-triangular part of $\mathbf{E}$ and then symmetrize the matrix.

The second component of the DiGress model is the denoising neural network $\phi_\theta$ parametrized by $\theta$. It takes a noisy graph $G^t = (\boldsymbol{X}^t, \mathbf{E}^t)$ as input and aims to predict the clean graph $G$, as illustrated in Figure 1. To train $\phi_\theta$, we optimize the cross-entropy loss $l$ between the predicted probabilities $\hat{p}^G = (\hat{p}^X, \hat{p}^E)$ for each node and edge and the true graph $G$:

$$l(\hat{p}^G, G) = \sum_{1 \leq i \leq n} \text{cross-entropy}(x_i, \hat{p}_i^X) + \lambda \sum_{1 \leq i,j \leq n} \text{cross-entropy}(e_{ij}, \hat{p}_{ij}^E) \tag{3}$$

where $\lambda \in \mathbb{R}^+$ controls the relative importance of nodes and edges. It is noteworthy that, unlike architectures like VAEs which solve complex distribution learning problems that sometimes requires graph matching, our diffusion model simply solves classification tasks on each node and edge.

Once the network is trained, it can be used to sample new graphs. To do so, we need to estimate the reverse diffusion iterations $p_\theta(G^{t-1}|G^t)$ using the network prediction $\hat{p}^G$. We model this distribu-

tion as a product over nodes and edges:

$$p_\theta(G^{t-1}|G^t) = \prod_{1 \le i \le n} p_\theta(x_i^{t-1}|G^t) \prod_{1 \le i,j \le n} p_\theta(e_{ij}^{t-1}|G^t) \tag{4}$$

To compute each term, we marginalize over the network predictions:

$$p_\theta(x_i^{t-1}|G^t) = \int_{x_i} p_\theta(x_i^{t-1} \mid x_i, G^t) \, dp_\theta(x_i|G^t) = \sum_{x \in \mathcal{X}} p_\theta(x_i^{t-1} \mid x_i = x, G^t) \, \hat{p}_i^X(x)$$

where we choose

$$p_\theta(x_i^{t-1} \mid x_i = x, \ G^t) = \begin{cases} q(x_i^{t-1} \mid x_i = x, \ x_i^t) & \text{if } q(x_i^t|x_i = x) > 0 \\ 0 & \text{otherwise.} \end{cases} \tag{5}$$

Similarly, we have $p_\theta(e_{ij}^{t-1}|e_{ij}^t) = \sum_{e \in \mathcal{E}} p_\theta(e_{ij}^{t-1} \mid e_{ij} = e, e_{ij}^t) \, \hat{p}_{ij}^E(e)$. These distributions are used to sample a discrete $G^{t-1}$ that will be the input of the denoising network at the next time step. These equations can also be used to compute an evidence lower bound on the likelihood, which allows for easy comparison between models. The computations are provided in Appendix C.

## 3.2 DENOISING NETWORK PARAMETRIZATION

The denoising network takes a noisy graph $G^t = (\boldsymbol{X}, \mathsf{E})$ as input and outputs tensors $\boldsymbol{X}'$ and $\mathsf{E}'$ which represent the predicted distribution over clean graphs. To efficiently store information, our layers also manipulate graph-level features $\boldsymbol{y}$. We chose to extend the graph transformer network proposed by Dwivedi & Bresson (2021), as attention mechanisms are a natural model for edge prediction. Our model is described in details in Appendix B.1. At a high-level, it first updates node features using self-attention, incorporating edge features and global features using FiLM layers (Perez et al., 2018). The edge features are then updated using the unnormalized attention scores, and the graph-level features using pooled node and edge features. Our transformer layers also feature residual connections and layer normalization. To incorporate time information, we normalize the timestep to $[0, 1]$ and treat it as a global feature inside $\boldsymbol{y}$. The overall memory and time complexity of our network is $\Theta(n^2)$ per layer, due to the attention scores and the predictions for each edge.

## 3.3 EQUIVARIANCE PROPERTIES

Graphs are invariant to reorderings of their nodes, meaning that $n!$ matrices can represent the same graph. To efficiently learn from these data, it is crucial to devise methods that do not require augmenting the data with random permutations. This implies that gradient updates should not change if the train data is permuted. To achieve this property, two components are needed: a permutation equivariant architecture and a permutation invariant loss. DiGress satisfies both properties.

**Lemma 3.1.** *(Equivariance) DiGress is permutation equivariant.*

**Lemma 3.2.** *(Invariant loss) Any loss function (such as the cross-entropy loss of Eq. (3)) that can be decomposed as $\sum_i l_X(\hat{p}_i^X, x_i) + \sum_{i,j} l_E(\hat{p}_{ij}^E, e_{ij})$ for two functions $l_X$ and $l_E$ computed respectively on each node and each edge is permutation invariant.*

Lemma 3.2 shows that our model does not require matching the predicted and target graphs, which would be difficult and costly. This is because the diffusion process keeps track of the identity of the nodes at each step, it can also be interpreted as a physical process where points are distinguishable.

Equivariance is however not a sufficient for likelihood computation: in general, the likelihood of a graph is the sum of the likelihood of all its permutations, which is intractable. To avoid this computation, we can make sure that the generated distribution is exchangeable, i.e., that all permutations of generated graphs are equally likely (Köhler et al., 2020).

**Lemma 3.3.** *(Exchangeability)*
*DiGress yields exchangeable distributions, i.e., it generates graphs with node features $\boldsymbol{X}$ and adjacency matrix $\boldsymbol{A}$ that satisfy $\mathbb{P}(\boldsymbol{X}, \boldsymbol{A}) = P(\pi^T \boldsymbol{X}, \pi^T \boldsymbol{A} \pi)$ for any permutation $\pi$.*

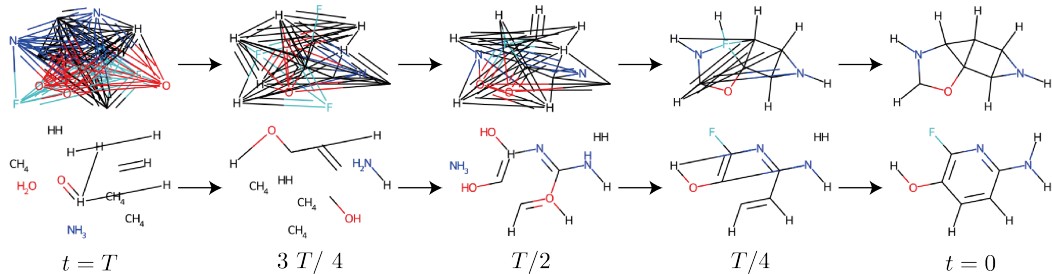

$t = T$ $\qquad$ $3\,T/\,4$ $\qquad$ $T/2$ $\qquad$ $T/4$ $\qquad$ $t = 0$

Figure 2: Reverse diffusion chains generated from a model trained on uniform transition noise (top) and marginal noise (bottom). When noisy graphs have the right marginals of node and edge types, they are closer to realistic graphs, which makes training easier.

## 4 Improving DiGress with marginal probabilities and structural features

### 4.1 Choice of the noise model

The choice of the Markov transition matrices $(\boldsymbol{Q}_t)_{t \leq T}$ defining the graph edit probabilities is arbitrary, and it is a priori not clear what noise model will lead to the best performance. A common model is a uniform transition over the classes $\boldsymbol{Q}^t = \alpha^t \boldsymbol{I} + (1 - \alpha^t)(\mathbf{1}_d \mathbf{1}_d')/d$, which leads to limit distributions $q_X$ and $q_E$ that are uniform over categories. Graphs are however usually sparse, meaning that the marginal distribution of edge types is far from uniform. Starting from uniform noise, we observe in Figure 2 that it takes many diffusion steps for the model to produce a sparse graph. To improve upon uniform transitions, we propose the following hypothesis: *using a prior distribution which is close to the true data distribution makes training easier.*

This prior distribution cannot be chosen arbitrarily, as it needs to be permutation invariant to satisfy exchangeability (Lemma 3.3). A natural model for this distribution is therefore a product $\prod_i u \times \prod_{i,j} v$ of a single distribution $u$ for all nodes and a single distribution $v$ for all edges. We propose the following result (proved in Appendix D) to guide the choice of $u$ and $v$:

**Theorem 4.1.** *(Optimal prior distribution)*
*Consider the class $\mathcal{C} = \{\prod_i u \times \prod_{i,j} v, \ (u, v) \in \mathcal{P}(\mathcal{X}) \times P(\mathcal{E})\}$ of distributions over graphs that factorize as the product of a single distribution $u$ over $\mathcal{X}$ for the nodes and a single distribution $v$ over $\mathcal{E}$ for the edges. Let $P$ be an arbitrary distribution over graphs (seen as a tensor of order $n + n^2$) and $m_X, m_E$ its marginal distributions of node and edge types. Then $\pi^G = \prod_i m_X \times \prod_{i,j} m_E$ is the orthogonal projection of $P$ on $\mathcal{C}$:*

$$\pi^G \in \underset{(u,v) \in \mathcal{C}}{\arg\min} \ || \ P - \prod_{1 \leq i \leq n} u \times \prod_{1 \leq i,j \leq n} v ||_2^2$$

This result means that to get a prior distribution $q_X \times q_E$ close to the true data distribution, we should define transition matrices such that $\forall i, \ \lim_{T \to \infty} \bar{\boldsymbol{Q}}_X^T \mathbb{1}_i = \boldsymbol{m}_X$ (and similarly for edges). To achieve this property, we propose to use

$$\boldsymbol{Q}_X^t = \alpha^t \boldsymbol{I} + \beta^t \ \mathbf{1}_a \boldsymbol{m}_X' \quad \text{and} \quad \boldsymbol{Q}_E^t = \alpha^t \boldsymbol{I} + \beta^t \ \mathbf{1}_b \boldsymbol{m}_E' \tag{6}$$

With this model, the probability of jumping from a state $i$ to a state $j$ is proportional to the marginal probability of category $j$ in the training set. Since $(\mathbf{1}\boldsymbol{m}')^2 = \mathbf{1}\boldsymbol{m}'$, we still have $\bar{\boldsymbol{Q}}^t = \bar{\alpha}^t \boldsymbol{I} + \bar{\beta}^t \mathbf{1}\boldsymbol{m}'$ for $\bar{\alpha}^t = \prod_{\tau=1}^t \alpha^\tau$ and $\bar{\beta}^t = \prod_{\tau=1}^t (1 - \alpha^\tau)$. We follow the popular cosine schedule $\bar{\alpha}^t = \cos(0.5\pi(t/T + s)/(1 + s))^2$ with a small $s$. Experimentally, these marginal transitions improves over uniform transitions (Appendix F).

### 4.2 Structural features augmentation

Generative models for graphs inherit the limitations of graph neural networks, and in particular their limited representation power (Xu et al., 2019; Morris et al., 2019). One example of this limitation is the difficulty for standard message passing networks (MPNNs) to detect simple substructures

---

**Algorithm 1:** Training DiGress

---

**Input:** A graph $G = (\boldsymbol{X}, \boldsymbol{E})$
Sample $t \sim \mathcal{U}(1, ..., T)$
Sample $G^t \sim \boldsymbol{X}\bar{\boldsymbol{Q}}_X^t \times \boldsymbol{E}\bar{\boldsymbol{Q}}_E^t$     ▷ Sample a (discrete) noisy graph
$z \leftarrow f(G^t, t)$     ▷ Structural and spectral features
$\hat{p}^X, \hat{p}^E \leftarrow \phi_\theta(G^t, z)$     ▷ Forward pass
optimizer. step$(l_{CE}(\hat{p}^X, \boldsymbol{X}) + \lambda\, l_{CE}(\hat{p}^E, \boldsymbol{\mathsf{E}}))$     ▷ Cross-entropy

---

**Algorithm 2:** Sampling from DiGress

---

Sample $n$ from the training data distribution
Sample $G^T \sim q_X(n) \times q_E(n)$     ▷ Random graph
**for** $t = T$ **to** $1$ **do**
 $z \leftarrow f(G^t, t)$     ▷ Structural and spectral features
 $\hat{p}^X, \hat{p}^E \leftarrow \phi_\theta(G^t, z)$     ▷ Forward pass
 $p_\theta(x_i^{t-1}|G^t) \leftarrow \sum_x q(x_i^{t-1}|x_i = x, x_i^t)\, \hat{p}_i^X(x) \quad i \in 1, ..., n$    ▷ Posterior
 $p_\theta(e_{ij}^{t-1}|G^t) \leftarrow \sum_e q(e_{ij}^{t-1}|e_{ij} = e, e_{ij}^t)\, \hat{p}_{ij}^E(e) \quad i, j \in 1, ..., n$
 $G^{t-1} \sim \prod_i p_\theta(x_i^{t-1}|G^t) \prod_{ij} p_\theta(e_{ij}^{t-1}|G^t)$     ▷ Categorical distr.
**end**
**return** $G^0$

---

such as cycles (Chen et al., 2020), which raises concerns about their ability to accurately capture the properties of the data distribution. While more powerful networks have been proposed such as (Maron et al., 2019; Vignac et al., 2020; Morris et al., 2022), they are significantly more costly and slower to train. Another strategy to overcome this limitation is to augment standard MPNNs with features that they cannot compute on their own. For example, Bouritsas et al. (2022) proposed adding counts of substructures of interest, and Beaini et al. (2021) proposed adding spectral features, which are known to capture important properties of graphs (Chung & Graham, 1997).

DiGress operates on a discrete space and its noisy graphs are not complete, allowing for the computation of various graph descriptors at each diffusion step. These descriptors can be input to the network to aid in the denoising process, resulting in Algorithms 1 and 2 for training DiGress and sampling from it. The inclusion of these additional features experimentally improves performance, but they are not required for building a good model. The choice of which features to include and the computational complexity of their calculation should be considered, especially for larger graphs. The details of the features used in our experiments can be found in the Appendix B.1.

## 5   CONDITIONAL GENERATION

While good unconditional generation is a prerequisite, the ability to condition generation on graph-level properties is crucial for many applications. For example, in drug design, molecules that are easy to synthesize and have high activity on specific targets are of particular interest. One way to perform conditional generation is to train the denoising network using the target properties (Hoogeboom et al., 2022), but it requires to retrain the model when the conditioning properties changes.

To overcome this limitation, we propose a new discrete guidance scheme inspired by the classifier guidance algorithm (Sohl-Dickstein et al., 2015). Our method uses a regressor $g_\eta$ which is trained to predict target properties $\boldsymbol{y}_G$ of a clean graph $G$ from a noisy version of $G$: $g_\eta(G^t) = \hat{\boldsymbol{y}}$. This regressor guides the unconditional diffusion model $\phi_\theta$ by modulating the predicted distribution at each sampling step and pushing it towards graphs with the desired properties. The equations for the conditional denoising process are given by the following lemma:

**Lemma 5.1.** *(Conditional reverse noising process) (Dhariwal & Nichol, 2021)*
*Denote $\dot{q}$ the noising process conditioned on $\boldsymbol{y}_G$, $q$ the unconditional noising process, and assume that $\dot{q}(G^t|G, \boldsymbol{y}_G) = \dot{q}(G^t|G)$. Then we have $\dot{q}(G^{t-1}|G^t, \boldsymbol{y}_G) \propto q(G^{t-1}|G^t)\, \dot{q}(\boldsymbol{y}_G|G^{t-1})$.*

While we would like to estimate $q(G^{t-1}|G^t)\, \dot{q}(\boldsymbol{y}_G|G^{t-1})$ by $p_\theta(G^{t-1}|G^t)\, p_\eta(\boldsymbol{y}_G|G^{t-1})$, $p_\eta$ cannot be evaluated for all possible values of $G^{t-1}$. To overcome this issue, we view $G$ as a continuous

---

**Algorithm 3:** Sampling from DiGress with discrete regressor guidance.

---

**Input:** Unconditional model $\phi_\theta$, property regressor $g$, target $\boldsymbol{y}$, guidance scale $\lambda$, graph size $n$

Sample $G^T \sim q_X(n) \times q_E(n)$                 ▷ `Random graph`

**for** $t = T$ **to** $1$ **do**

     $z \leftarrow f(G^t, t)$               ▷ `Structural and spectral features`

     $\hat{p}^X, \hat{p}^E \leftarrow \phi_\theta(G^t, z)$            ▷ `Forward pass`

     $\hat{\boldsymbol{y}} \leftarrow g_\eta(G^t)$              ▷ `Regressor model`

     $p_\eta(\hat{\boldsymbol{y}}|G^{t-1}) \propto \exp(-\lambda \langle \nabla_{G^t} ||\hat{\boldsymbol{y}} - \boldsymbol{y}||^2, G^{t-1}\rangle)$     ▷ `Guidance distribution`

     Sample $G^{t-1} \sim p_\theta(G^{t-1}|G^t)\, p_\eta(\hat{\boldsymbol{y}}|G^{t-1})$       ▷ `Reverse process`

**end**

**return** $G^0$

---

tensor of order $n + n^2$ (so that $\nabla_G$ can be defined) and use a first-order approximation. It gives:

$$\log \dot{q}(\boldsymbol{y}_G|G^{t-1}) \approx \log \dot{q}(\boldsymbol{y}_G|G^t) + \langle \nabla_G \log \dot{q}(\boldsymbol{y}_G|G^t), G^{t-1} - G^t\rangle$$
$$\approx c(G^t) + \sum_{1 \leq i \leq n} \langle \nabla_{x_i} \log \dot{q}(\boldsymbol{y}_G|G^t), \boldsymbol{x}_i^{t-1}\rangle + \sum_{1 \leq i,j \leq n} \langle \nabla_{e_{ij}} \log \dot{q}(\boldsymbol{y}_G|G^t), \boldsymbol{e}_{ij}^{t-1}\rangle$$

for a function $c$ that does not depend on $G^{t-1}$. We make the additional assumption that $\dot{q}(\boldsymbol{y}_G|G^t) = \mathcal{N}(g(G^t), \sigma_y \boldsymbol{I})$, where $g$ is estimated by $g_\eta$, so that $\nabla_{G^t} \log \dot{q}_\eta(\boldsymbol{y}|G^t) \propto -\nabla_{G^t}||\hat{\boldsymbol{y}} - \boldsymbol{y}_G||^2$. The resulting procedure is presented in Algorithm 3.

In addition to being conditioned on graph-level properties, our model can be used to extend an existing subgraph – a task called molecular scaffold extension in the drug discovery literature (Maziarz et al., 2022). In Appendix E, we explain how to do it and demonstrate it on a simple example.

## 6   RELATED WORK

Several works have recently proposed discrete diffusion models for text, images, audio, and attributed point clouds (Austin et al., 2021; Yang et al., 2022; Luo et al., 2022). Our work, DiGress, is the first discrete diffusion model for graphs. Concurrently, Haefeli et al. (2022) designed a model limited to unattributed graphs, and similarly observed that discrete diffusion is beneficial for graph generation. Previous diffusion models for graphs were based on Gaussian noise: Niu et al. (2020) generated adjacency matrices by thresholding a continuous value to indicate edges, and Jo et al. (2022) extended this model to handle node and edge attributes.

Trippe et al. (2022), Hoogeboom et al. (2022) and Wu et al. (2022) define diffusion models for molecule generation in 3D. These models actually solve a point cloud generation task, as they generate atomic positions rather than graph structures and thus require conformer data for training. On the contrary, Xu et al. (2022) and Jing et al. (2022) define diffusion model for conformation generation – they input a graph structure and output atomic coordinates.

Apart from diffusion models, there has recently been a lot of interest in non-autoregressive graph generation using VAEs, GANs or normalizing flows (Zhu et al., 2022). (Madhawa et al., 2019; Lippe & Gavves, 2021; Luo et al., 2021) are examples of discrete models using categorical normalizing flows. However, these methods have not yet matched the performance of autoregressive models (Liu et al., 2018; Liao et al., 2019; Mercado et al., 2021) and motifs-based models (Jin et al., 2020; Maziarz et al., 2022), which can incorporate much more domain knowledge.

## 7   EXPERIMENTS

In our experiments, we compare the performance of DiGress against several state-of-the-art one-shot graph generation methods on both molecular and non-molecular benchmarks. We compare its performance against Set2GraphVAE (Vignac & Frossard, 2021), SPECTRE (Martinkus et al., 2022), GraphNVP (Madhawa et al., 2019), GDSS (Jo et al., 2022), GraphRNN (You et al., 2018), GRAN (Liao et al., 2019), JT-VAE (Jin et al., 2018), NAGVAE (Kwon et al., 2020) and GraphINVENT

Table 2: Molecule generation on QM9. Training time is the time needed to reach 99% validity. On small graphs, DiGress achieves similar results to the continuous model but is faster to train.

| Method | NLL | Valid | Unique | Training time (h) |
|---|---|---|---|---|
| Dataset | – | 99.3 | 100 | – |
| Set2GraphVAE | – | 59.9 | 93.8 | – |
| SPECTRE | – | 87.3 | 35.7 | – |
| GraphNVP | – | 83.1 | 99.2 | – |
| GDSS | – | 95.7 | **98.5** | – |
| ConGress (ours) | – | 98.9$_{\pm.1}$ | 96.8$_{\pm.2}$ | 7.2 |
| DiGress (ours) | 69.6$_{\pm1.5}$ | **99.0**$_{\pm.1}$ | 96.2$_{\pm.1}$ | **1.0** |

(Mercado et al., 2021). We also build Congress, a model that has the same denoising network as DiGress but Gaussian diffusion (Appendix A). Our results are presented without validity correction[1].

### 7.1 GENERAL GRAPH GENERATION

Table 1: Unconditional generation on SBM and planar graphs. VUN: valid, unique & novel graphs.

| Model | Deg ↓ | Clus ↓ | Orb↓ | V.U.N. ↑ |
|---|---|---|---|---|
| *Stochastic block model* | | | | |
| GraphRNN | 6.9 | 1.7 | 3.1 | 5 % |
| GRAN | 14.1 | 1.7 | 2.1 | 25% |
| GG-GAN | 4.4 | 2.1 | 2.3 | 25% |
| SPECTRE | 1.9 | 1.6 | **1.6** | 53% |
| ConGress | 34.1 | 3.1 | 4.5 | 0% |
| DiGress | **1.6** | **1.5** | 1.7 | **74**% |
| *Planar graphs* | | | | |
| GraphRNN | 24.5 | 9.0 | 2508 | 0% |
| GRAN | 3.5 | 1.4 | 1.8 | 0% |
| SPECTRE | 2.5 | 2.5 | 2.4 | 25% |
| ConGress | 23.8 | 8.8 | 2590 | 0% |
| DiGress | **1.4** | **1.2** | 1.7 | **75**% |

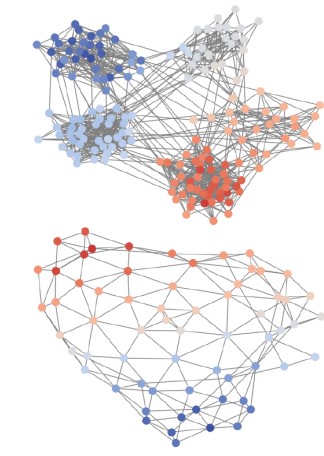

Figure 3: Samples from DiGress trained on SBM and planar graphs.

We first evaluate DiGress on the benchmark proposed in Martinkus et al. (2022), which consists of two datasets of 200 graphs each: one drawn from the stochastic block model (with up to 200 nodes per graph), and another dataset of planar graphs (64 nodes per graph). We evaluate the ability to correctly model various properties of these graphs, such as whether the generated graphs are statistically distinguishable from the SBM model or if they are planar and connected. We refer to Appendix F for a description of the metrics. In Table 1, we observe that DiGress is able to capture the data distribution very effectively, with significant improvements over baselines on planar graphs. In contrast, our continuous model, ConGress, performs poorly on these relatively large graphs.

### 7.2 SMALL MOLECULE GENERATION

We then evaluate our model on the standard QM9 dataset (Wu et al., 2018) that contains molecules with up to 9 heavy atoms. We use a split of 100k molecules for training, 20k for validation and 13k for evaluating likelihood on a test set. We report the negative log-likelihood of our model, validity (measured by RDKit sanitization) and uniqueness over 10k molecules. Novelty results are discussed in Appendix F. 95% confidence intervals are reported based on five runs. Results are presented in Figure 2. Since ConGress and DiGress both obtain close to perfect metrics on this dataset, we also perform an ablation study on a more challenging version of QM9 where hydrogens are explicitly modeled in Appendix F. It shows that the discrete framework is beneficial and that marginal transitions and auxiliary features further boost performance.

---

[1]Code is available at `github.com/cvignac/DiGress`.

Table 3: Molecule generation on MOSES. DiGress is the first one-shot graph model that scales to this dataset. While all graph-based methods except ours have hard-coded rules to ensure high validity, DiGress outperforms GraphInvent on most other metrics.

| Model | Class | Val ↑ | Unique↑ | Novel↑ | Filters↑ | FCD↓ | SNN↑ | Scaf↑ |
|---|---|---|---|---|---|---|---|---|
| VAE | SMILES | 97.7 | 99.8 | 69.5 | 99.7 | 0.57 | 0.58 | 5.9 |
| JT-VAE | Fragment | 100 | 100 | 99.9 | 97.8 | 1.00 | 0.53 | 10 |
| GraphINVENT | Autoreg. | 96.4 | 99.8 | – | 95.0 | 1.22 | 0.54 | 12.7 |
| ConGress (ours) | One-shot | 83.4 | 99.9 | 96.4 | 94.8 | 1.48 | 0.50 | 16.4 |
| DiGress (ours) | One-shot | 85.7 | 100 | 95.0 | 97.1 | 1.19 | 0.52 | 14.8 |

Table 4: Molecule generation on GuacaMol. We report scores, so that higher is better for all metrics. While SMILES seem to be the most efficient molecular representation, DiGress is the first general graph generation method that achieves correct performance, as visible on the FCD score.

| Model | Class | Valid↑ | Unique↑ | Novel↑ | KL div↑ | FCD↑ |
|---|---|---|---|---|---|---|
| LSTM | Smiles | 95.9 | 100 | 91.2 | 99.1 | 91.3 |
| NAGVAE | One-shot | 92.9 | 95.5 | 100 | 38.4 | 0.9 |
| MCTS | One-shot | 100 | 100 | 95.4 | 82.2 | 1.5 |
| ConGress (ours) | One-shot | 0.1 | 100 | 100 | 36.1 | 0.0 |
| DiGress (ours) | One-shot | 85.2 | 100 | 99.9 | 92.9 | 68.0 |

## 7.3 CONDITIONAL GENERATION

To measure the ability of DiGress to condition the generation on graph-level properties, we propose a conditional generation setting on QM9. We sample 100 molecules from the test set and retrieve their dipole moment $\mu$ and the highest occupied molecular orbit (HOMO).

Figure 4: Mean absolute error on conditional generation with discrete regression guidance on QM9.

| Target | $\mu$ | HOMO | $\mu$ & HOMO |
|---|---|---|---|
| Uncondit. | $1.71_{\pm.04}$ | $0.93_{\pm.01}$ | $1.34_{\pm.01}$ |
| Guidance | $0.81_{\pm.04}$ | $0.56_{\pm.01}$ | $0.87_{\pm.03}$ |

The pairs $(\mu, \text{HOMO})$ constitute the conditioning vector that we use to generate 10 molecules. To evaluate the ability of a model to condition correctly, we need to estimate the properties of the generated samples. To do so, we use RdKit (Landrum et al., 2006) to produce conformers of the generated graphs, and then Psi4 (Smith et al., 2020) to estimate the values of $\mu$ and HOMO. We report the mean absolute error between the targets and the estimated values for the generated molecules (Fig. 4).

## 7.4 MOLECULE GENERATION AT SCALE

We finally evaluate our model on two much more challenging datasets made of more than a million molecules: MOSES (Polykovskiy et al., 2020), which contains small drug-like molecules, and GuacaMol (Brown et al., 2019), which contains larger molecules. DiGress is to our knowledge the first one-shot generative model that is not based on molecular fragments and that scales to datasets of this size. The metrics used as well as additional experiments are presented in App. F. For MOSES, the reported scores for FCD, SNN, and Scaffold similarity are computed on the dataset made of separate scaffolds, which measures the ability of the networks to predict new ring structures. Results are presented in Tables 3 and 4: they show that DiGress does not yet match the performance of SMILES and fragment-based methods, but performs on par with GraphInvent, an autoregressive model fine-tuned using chemical softwares and reinforcement learning. DiGress thus bridges the important gap between one-shot methods and autoregressive models that previously prevailed.

## 8 CONCLUSION

We proposed DiGress, a denoising diffusion model for graph generation that operates on a discrete space. DiGress outperforms existing one-shot generation methods and scales to larger molecular datasets, reaching the performance of autogressive models trained using expert knowledge.

## ACKNOWLEDGMENTS

We thank Nikos Karalias, Éloi Alonso and Karolis Martinkus for their help and useful suggestions. Clément Vignac thanks the Swiss Data Science Center for supporting him through the PhD fellowship program (grant P18-11). Igor Krawczuk and Volkan Cevher acknowledge funding from the European Research Council (ERC) under the European Union's Horizon 2020 research and innovation programme (grant agreement n°725594 - time-data).

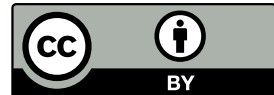

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

# A  CONTINUOUS GRAPH DENOISING DIFFUSION MODEL (CONGRESS)

In this section we present a diffusion model for graphs that uses Gaussian noise rather than a discrete diffusion process. Its denoising network is the same as the one of our discrete model. Our goal is to show that the better performance obtained with DiGress is not only due to the neural network design, but also to the discrete process itself.

## A.1  DIFFUSION PROCESS

Consider a graph $G = (\boldsymbol{X}, \mathbf{E})$. Similarly to the discrete diffusion model, this diffusion process adds noise independently on each node and each edge, but this time the noise considered is Gaussian:

$$q(\boldsymbol{X}^t|\boldsymbol{X}^{t-1}) = \mathcal{N}(\alpha^{t|t-1}\boldsymbol{X}^{t-1}, (\sigma^{t|t-1})^2\mathbf{I}) \quad \text{and} \quad q(\mathbf{E}^t|\mathbf{E}^{t-1}) = \mathcal{N}(\alpha^{t|t-1}\mathbf{E}^{t-1}, (\sigma^{t|t-1})^2\mathbf{I}) \tag{7}$$

This process can equivalently be written:

$$q(\boldsymbol{X}^t|\boldsymbol{X}) = \mathcal{N}(\boldsymbol{X}^t|\alpha^t\boldsymbol{X}, \sigma^t\boldsymbol{I}) \qquad q(\mathbf{E}^t|\mathbf{E}) = \mathcal{N}(\mathbf{E}^t|\alpha^t\mathbf{E}, \sigma^t\mathbf{I}) \tag{8}$$

where $\alpha^{t|t-1} = \alpha^t/\alpha^{t-1}$ and $(\sigma^{t|t-1})^2 = (\sigma^t)^2 - (\alpha^{t|t-1})^2(\sigma^{t-1})^2$.

The variance is chosen as $(\sigma^t)^2 = 1 - (\alpha^t)^2$ in order to obtain a *variance-preserving* process (Kingma et al., 2021). Similarly to DiGress, when we consider undirected graphs, we only apply the noise on the upper-triangular part of $\mathbf{E}$ without the main diagonal, and then symmetrize the matrix. The true denoising process can be computed in closed-form:

$$q(\boldsymbol{X}^{t-1}|\boldsymbol{X}, \boldsymbol{X}^t) = \mathcal{N}(\boldsymbol{\mu}^{t\to t-1}(\boldsymbol{X}, \boldsymbol{X}^t), (\sigma^{t\to t-1})^2\,\mathbf{I}) \quad \text{(and similarly for } \mathbf{E}), \tag{9}$$

---

**Algorithm 4:** Training ConGress

---

**Input:** A graph $G = (\boldsymbol{X}, \mathbf{E})$
Sample $t \sim \mathcal{U}(1, ..., T)$
Sample $\epsilon_X \sim \mathcal{N}(0, \boldsymbol{I}_n)$
Sample $\epsilon_E \sim \mathcal{N}(0, \boldsymbol{I}_{n(n-1)/2})$ and symmetrize if needed
$z^t \leftarrow \alpha^t(\boldsymbol{X}, \mathbf{E}) + \sigma_t(\epsilon_X, \epsilon_E)$            ▷ Add noise
Minimize $||(\epsilon_X, \epsilon_E) - \phi_\theta(z^t, t)||^2$

---

**Algorithm 5:** Sampling from ConGress

---

Sample $n$ from the training data distribution
Sample $\epsilon_X \sim \mathcal{N}(0, \boldsymbol{I}_n)$
Sample $\epsilon_E \sim \mathcal{N}(0, \boldsymbol{I}_{n(n-1)/2})$ and symmetrize if needed
$z_T \leftarrow (\epsilon_X, \epsilon_E)$
**for** $t = T$ **to** $1$ **do**
     Sample $\epsilon_X \sim \mathcal{N}(0, \boldsymbol{I}_n)$
     Sample and symmetrize $\epsilon_E \sim \mathcal{N}(0, \boldsymbol{I}_{n(n-1)/2})$
     $z^{t-1} \leftarrow \frac{1}{\alpha_{t|t-1}} z^t - \frac{\sigma^2_{t|t-1}}{\alpha_{t|t-1}\sigma^t} \phi_\theta(z^t, t) + \sigma_{t \to t-1}(\epsilon_X, \epsilon_E)$     ▷ Reverse iterations
**end**
**return** $\operatorname{argmax}(\boldsymbol{X}^0), \operatorname{argmax}(\mathbf{E}^0)$

---

with

$$\boldsymbol{\mu}^{t \to t-1}(\boldsymbol{X}, \boldsymbol{X}^t) = \frac{\alpha_{t|t-1}(\sigma^{t-1})^2}{\sigma_t^2}\boldsymbol{X}^t + \frac{\alpha^{t-1}(\sigma^{t|t-1})^2}{(\sigma^t)^2}\boldsymbol{X} \quad \text{and} \quad \sigma^{t \to t-1} = \frac{\sigma_{t|t-1}\,\sigma_{t-1}}{\sigma_t}. \quad (10)$$

As commonly done for Gaussian diffusion models, we train the denoising network to predict the noise components $\hat{\epsilon}_X, \hat{\epsilon}_E$ instead of $\hat{\boldsymbol{X}}$ and $\hat{\mathbf{E}}$ themselves (Ho et al., 2020). Both relate as follows:

$$\alpha^t \hat{\boldsymbol{X}} = \boldsymbol{X}^t - \sigma^t \hat{\boldsymbol{\epsilon}}_X \quad \text{and} \quad \hat{\alpha}^t \mathbf{E} = \mathbf{E}^t - \sigma^t \hat{\boldsymbol{\epsilon}}_E \quad (11)$$

To optimize the network, we minimize the mean squared error between the predicted noise and the true noise, which results in Algorithm 4 for training ConGress. Sampling is done similarly to standard Gaussian diffusion models, except for the last step: since continuous valued features are obtained, they must be mapped back to categorical values in order to obtain a discrete graph. For this purpose, we then take the argmax of $\boldsymbol{X}^0, \mathbf{E}^0$ across node and edge types (Algorithm 5).

Overall, ConGress is very close to the GDSS model proposed in Jo et al. (2022), as it is also a Gaussian-based diffusion model for graphs. An important difference is that we define a diffusion process that is independent for each node and edge, while GDSS uses a more complex noise model that does not factorize. We observe empirically that a simple noise model does not hurt performance, since ConGress outperforms GDSS on QM9 (Table 2).

# B  NEURAL NETWORK PARAMETRIZATION

## B.1  GRAPH TRANSFORMER NETWORK

The parametrization of our denoising network is presented in Figure 5. It takes as input a noisy graph $(\boldsymbol{X}, \mathbf{E})$ and predicts a distribution over the clean graphs. We compute structural and spectral features in order to improve the network expressivity. Internally, each layer manipulates nodes features $\boldsymbol{X}$, edge features $\mathbf{E}$ but also graph level features $\boldsymbol{y}$. Each graph transformer layer is made of a graph attention module (presented in Figure 6), as well as a fully-connected layers and layer normalization.

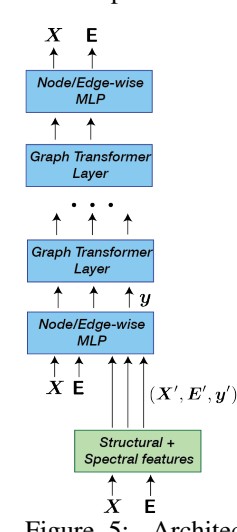

Figure 5: Architecture of the denoising

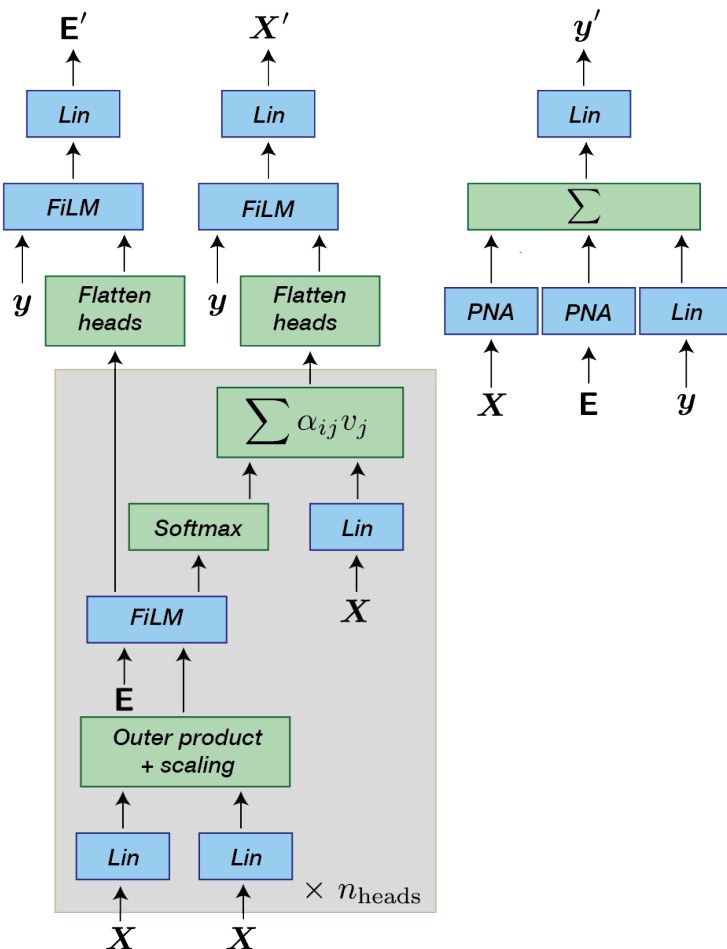

Figure 6: The self-attention module of our graph transformer network. It takes as input node features $\boldsymbol{X}$, edge features $\mathbf{E}$ and global features $\boldsymbol{y}$, and updates their representation. These features are then passed to normalization layers and a fully connected network, similarly to the standard transformer architecture. $\mathrm{FiLM}(\boldsymbol{M}_1, \boldsymbol{M}_2) = \boldsymbol{M}_1 \boldsymbol{W}_1 + (\boldsymbol{M}_1 \boldsymbol{W}_2) \odot \boldsymbol{M}_2 + \boldsymbol{M}_2$ for learnable weight matrices $\boldsymbol{W}_1$ and $\boldsymbol{W}_2$, and $\mathrm{PNA}(\boldsymbol{X}) = \mathrm{cat}(\max(\boldsymbol{X}), \min(\boldsymbol{X}), \mathrm{mean}(\boldsymbol{X}), \mathrm{std}(\boldsymbol{X})) \, \boldsymbol{W}$.

## B.2 AUXILIARY STRUCTURAL AND SPECTRAL FEATURES

The structural features that we use can be divided in two types: graph-theoretic (cycles and spectral features) and domain specific (molecular features).

**Cycles** Since message-passing neural networks are unable to detect cycles (Chen et al., 2020), we add cycle counts to our model. Because computing traversals would be impractical on GPUs (all the more as these features are recomputed at every diffusion step), we use formulas for cycles up to size 6. We build node features (how many $k-$cycles does this node belong to?) for up to 5-cycles, and graph-level features (how many $k-$cycles does this graph contain?) for up to $k = 6$. We use the following formulas, where $\boldsymbol{d}$ denotes the vector containing node degrees and $||.||_F$ is Frobenius

norm:

$$\boldsymbol{X}_3 = \mathrm{diag}(\boldsymbol{A}^3)/2$$

$$\boldsymbol{X}_4 = (\mathrm{diag}(\boldsymbol{A}^4) - \boldsymbol{d}(\boldsymbol{d}-1) - \boldsymbol{A}(\boldsymbol{d}\mathbf{1}_n^T)\mathbf{1}_n)/2$$

$$\boldsymbol{X}_5 = (\mathrm{diag}(\boldsymbol{A}^5) - 2\,\mathrm{diag}(\boldsymbol{A}^3) \odot \boldsymbol{d} - \boldsymbol{A}(\mathrm{diag}(\boldsymbol{A}^3)\mathbf{1}_n^T)\mathbf{1}_n + \mathrm{diag}(\boldsymbol{A}^3))/2$$

$$\boldsymbol{y}_3 = \boldsymbol{X}_3^T\mathbf{1}_n/3$$

$$\boldsymbol{y}_4 = \boldsymbol{X}_4^T\mathbf{1}_n/4$$

$$\boldsymbol{y}_5 = \boldsymbol{X}_5^T\mathbf{1}_n/5$$

$$\boldsymbol{y}_6 = \mathrm{Tr}(\boldsymbol{A}^6) - 3\,\mathrm{Tr}(\boldsymbol{A}^3 \odot \boldsymbol{A}^3) + 9||\boldsymbol{A}(\boldsymbol{A}^2 \odot \boldsymbol{A}^2)||_F - 6\,\langle\mathrm{diag}(A^2),\mathrm{diag}(A^4)\rangle$$
$$+ 6\,\mathrm{Tr}(\boldsymbol{A}^4) - 4\,\mathrm{Tr}(\boldsymbol{A}^3) + 4\,\mathrm{Tr}(\boldsymbol{A}^2\dot{\boldsymbol{A}}^2 \odot \boldsymbol{A}^2) + 3||\boldsymbol{A}^3||_F - 12\,\mathrm{Tr}(\boldsymbol{A}^2 \odot \boldsymbol{A}^2) + 4\,\mathrm{Tr}(\boldsymbol{A}^2)$$

**Spectral features** We also add the option to incorporate spectral features to the model. While this requires a $O(n^3)$ eigendecomposition, we find that it is not a limiting factor for the graphs that we use in our experiments (that have up to 200 nodes). We first compute some graph-level features that relate to the eigenvalues of the graph Laplacian: the number of connected components (given by the multiplicity of eigenvalue 0), as well as the 5 first nonzero eigenvalues. We then add node-level features relative to the graph eigenvectors: an estimation of the biggest connected component (using the eigenvectors associated to eigenvalue 0), as well as the two first eigenvectors associated to non zero eigenvalues.

**Molecular features** On molecular datasets, we also incorporate the current valency of each atom and the current molecular weight of the full molecule.

## C LIKELIHOOD COMPUTATION

Figure 7: The graphical model of DiGress and ConGress

The graphical model associated to our problem is presented in figure 7: the graph size is sampled from the training distribution and kept constant during diffusion. One can notice the similarity between this graphical model and hierarchical variational autoencoders (VAEs): diffusion models can in fact be interpreted as a particular instance of VAE where the encoder (i.e., the diffusion process) is fixed. The likelihood of a data point $x$ under the model writes:

$$\log p_\theta(G) = \log \sum_{n\in\mathbb{N}} p(n) \int p(G^T\,|\,n)\, p_\theta(G^{t-1},\dots,G^1|G^T)\, p_\theta(G|G^1)\, d(G^1,\dots,G^T) \quad (12)$$

$$= \log p(n_G) + \log \int p(G^T|\,n_G) \prod_{t=2}^{T} p_\theta(G^{t-1}\,|\,G^t)\, p_\theta(G|G^1)\, d(G^1,\dots,G^T) \quad (13)$$

As for VAEs, an evidence lower bound (ELBO) for this integral can be computed (Sohl-Dickstein et al., 2015; Kingma et al., 2021). It writes:

$$\log p_\theta(G) \geq \log p(n_G) + \underbrace{D_{\mathrm{KL}}[q(G^T|G)\,||\,q_X(n_G)\times q_E(n_G)]}_{\text{Prior loss}} + \underbrace{\sum_{t=2}^{T} L_t(x)}_{\text{Diffusion loss}} + \underbrace{\mathbb{E}_{q(G^1|G)}[\log p_\theta(G|G^1)]}_{\text{Reconstruction loss}}$$
$$(14)$$

with

$$L_t(G) = \mathbb{E}_{q(G^t|G)}\big[D_{\mathrm{KL}}[q(G^{t-1}|G^t,G)\,||\,p_\theta(G^{t-1}|G^t)]\big] \quad (15)$$

All these terms can be estimated: $\log p(n_G)$ is computed using the frequencies of the number of nodes for each graph in the dataset. The prior loss and the diffusion loss are KL divergences between categorical distribution, and the reconstruction loss is simply computed from the predicted probabilities for the clean graph given the last noisy graph $G^1$.

# D  PROOFS

**True posterior distribution**

We recall the derivation of the true posterior distribution $q(z^{t-1}|z^t, x) \propto \boldsymbol{z}^t\,(\boldsymbol{Q}^t)' \odot \boldsymbol{x}\,\bar{\boldsymbol{Q}}^{t-1}$.

By Bayes rule, we have:

$$q(z^{t-1}|z^t, x) \propto q(z^t|z^{t-1}, x)\,q(z^{t-1}|x)$$

Since the noise is Markovian, $q(z^t|z^{t-1}, x) = q(z^t|z^{t-1})$. A second application of Bayes rule gives $q(z^t|z^{t-1}) \propto q(z^{t-1}|z^t)q(z^t)$.

By writing the definition of $\boldsymbol{Q}^t$, we then observe that $q(z^{t-1}|z^t) = \boldsymbol{z}^t\,(\boldsymbol{Q}^t)'$. We also have $q(z^{t-1}|x) = \boldsymbol{x}\,\bar{\boldsymbol{Q}}^{t-1}$ by definition.

Finally, we observe that $q(z^t)$ does not depend on $z^{t-1}$. It can therefore be seen as a part of the normalization constant. By combining the terms, we have $q(z^{t-1}|z^t, x) \propto \boldsymbol{z}^t\,(\boldsymbol{Q}^t)' \odot \boldsymbol{x}\,\bar{\boldsymbol{Q}}^{t-1}$ as desired.

**Lemma 3.1: Equivariance**

*Proof.* Consider a graph $G$ with $n$ nodes, and $\pi \in S_n$ a permutation. $\pi$ acts trivially on $\boldsymbol{y}$ ($\pi.\boldsymbol{y} = \boldsymbol{y}$), it acts on $\boldsymbol{X}$ as $\pi.\boldsymbol{X} = \pi'X$ and on $\mathbf{E}$ as:

$$(\pi.\mathbf{E})_{ijk} = \mathbf{E}_{\pi^{-1}(i),\pi^{-1}(j),k}$$

Let $G^t = (\boldsymbol{X}^t, \mathbf{E}^t)$ be a noised graph, and $(\pi.\boldsymbol{X}^t, \pi.\mathbf{E}^t)$ its permutation. Then:

- Our spectral and structural features are all permutation equivariant (for the node features) or invariant (for the graph level features): $f(\pi.G^t, t) = \pi.f(G^t, t)$.

- The self-attention architecture is permutation equivariant. The FiLM blocks are permutation equivariant, and the PNA pooling function is permutation invariant.

- Layer-normalization is permutation equivariant.

DiGress is therefore the combination of permutation equivariant blocks. As a result, it is permutation equivariant: $\phi_\theta(\pi.G^t, f(\pi.G^t, t)) = \pi.\phi_\theta(G^t, f(G^t, t))$. □

**Lemma 3.2: Invariant loss**

*Proof.* It is important that the loss function be the same for each node and each edge in order to guarantee that

$$l(\pi.\hat{G}, \pi.G) = \sum_i l_X(\pi.\hat{\boldsymbol{X}}_i, x_{\pi^{-1}(i)}) + \sum_{i,j} l_E(\pi.\hat{\mathbf{E}}_{ij}, e_{\pi^{-1}(i),\pi^{-1}(j)})$$

$$= \sum_i l_X(\hat{\boldsymbol{X}}_i, x_i) + \sum_{i,j} l_E(\hat{\mathbf{E}}_{ij}, e_{i,j})$$

$$= l(\hat{G}, G)$$

□

**Lemma 3.3: Exchangeability**

*Proof.* The proof relies on the result of Xu et al. (2022): if a distribution $p(G^T)$ is invariant to the action of a group $\mathcal{G}$ and the transition probabilities $p(G^{t-1}|G^t)$ are equivariant, them $p(G^0)$ is invariant to the action of $\mathcal{G}$. We apply this result to the special case of permutations:

- The limit noise distribution is the product of i.i.d. distributions on each node and edge. It is therefore permutation invariant.
- The denoising neural networks is permutation equivariant.
- The function $\hat{p}_\theta(G) \to p_\theta(G^{t-1}|G^t) = \sum_G q(G^{t-1}, G|G^t)\hat{p}_\theta(G)$ defining the transition probabilities is equivariant to joint permutations of $\hat{p}_\theta(G)$ and $G^t$.

The conditions of (Xu et al., 2022) are therefore satisfied, and the model satisfies $\mathbb{P}(\boldsymbol{X}, \mathbf{E}) = P(\pi.\boldsymbol{X}, \pi.\mathbf{E})$ for any permutation $\pi$. □

**Theorem 4.1: Optimal prior distribution**   We first prove the following result:

**Lemma D.1.** *Let $p$ be a discrete distribution over two variables. It is represented by a matrix $P \in \mathbb{R}^{a \times b}$. Let $m^1$ and $m^2$ the marginal distribution of $p$: $m_i^1 = \sum_{j=1}^{b} p_{ij}$ and $m_i^2 = \sum_{i=1}^{a} p_{ij}$. Then*

$$(m^1, m^2) \in \operatorname*{arg\,min}_{\substack{u,v \\ u \geq 0, \sum u_i = 1 \\ v \geq 0, \sum v_j = 1}} ||P - u\, v'||_2^2$$

*Proof.* We define $L(\boldsymbol{u}, \boldsymbol{v}) := ||P - uv'||_2^2 = \sum_{i,j} (p_{ij} - u_i v_j)^2$. We derive this formula to obtain optimality conditions:

$$\frac{L}{\partial u_i} = 0 \iff \sum_j (p_{ij} - u_i v_j) v_i = 0$$

$$\iff \sum_j p_{ij} v_j = u_i \sum_j v_j^2$$

$$\iff u_i = \sum_j p_{ij} v_j \,/\, \sum_j v_j^2$$

Similarly, we have $\frac{\partial L}{\partial v_j} = 0 \iff v_j = \sum_i p_{ij} u_i \,/\, \sum_j u_i^2$.

Since $p$, $u$ and $v$ are probability distributions, we have $\sum_{i,j} p_{i,j} = 1$, $\sum_i u_i = 1$ and $\sum_j v_j = 1$. Combining these equations, we have:

$$u_i = \frac{\sum_j p_{ij} v_j}{\sum_j v_j^2} \implies \sum_i u_i = 1 = \frac{\sum_{i,j} p_{ij} v_j}{\sum_j v_j^2}$$

$$\iff \sum_j v_j^2 = \sum_j (\sum_i p_{ij}) v_j$$

$$\iff \sum_j v_j^2 = \sum_j b_j v_j$$

So that:

$$u_{i_0} = \frac{\sum_j p_{i_0 j} v_j}{\sum_j b_j v_j} = \frac{\sum_j p_{i_0 j} \frac{\sum_i p_{ij} u_i}{\sum_i a_i u_i}}{\sum_j b_j \frac{\sum_i p_{ij} u_i}{\sum_i a_i u_i}} = \sum_j p_{i_0 j} = m_{i_0}^1$$

and similarly $v_{j_0} = b_{i_0}$. Conversely, $\boldsymbol{m}^1$ and $\boldsymbol{m}^2$ belong to the set of feasible solutions.   $\square$

We have proved that the product distribution that is the closest to the true distribution of two variables is the product of marginals (for $l_2$ distance). We need to extend this result to a product $\prod_{i=1}^{n} u \times \prod_{1 \leq i,j \leq n} v$ of a distribution for nodes and a distribution for edges.

We now view $p$ as a tensor in dimension $a^n b^{n^2}$. We denote $p^X$ the marginalisation of this tensor across the node dimensions ($p^X \in \mathbb{R}^{a^n}$), and $p^E$ the marginalisation across the edge dimensions ($p^E \in \mathbb{R}^{b^{n \times n}}$). By flattening the $n$ first dimensions and the $n^2$ next, $p$ can be viewed as a distribution over two variables (a distribution for the nodes and a distribution for the edges). By application of our Lemma, $p^X$ and $p^E$ are the optimal approximation of $p$. However, $p^X$ is a joint distribution for all nodes and not the product $\prod_{i=1}^{n} u$ of a single distribution for all nodes.

We then notice that:

$$||\prod_{i=1}^{n} u - p^X||_2^2 = \sum_i ||u||^2 - 2 \sum_i \langle u, p_i^X \rangle + \sum_i ||p_i^X||^2$$

$$= n \left( ||u||^2 - 2 \langle u, \frac{1}{n} \sum_i p_i^X \rangle + \frac{1}{n} \sum_i ||p_i^X||^2 \right)$$

$$= ||u - \frac{1}{n} \sum_i p_i^X||_2^2 + f(p^X)$$

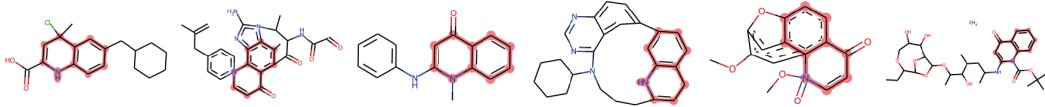

Figure 8: An example of molecular scaffold extension. We sometimes observe long-range consistency issues in the generated samples, which is in line with the observations of (Lugmayr et al., 2022) for image data. A resampling strategy similar to theirs could be used to solve this issue.

for a function $f$ that does not depend on $u$. As $\sum_i p_i^X / n$ is exactly the empirical distribution of node types, the optimal $u$ is the empirical distribution of node types as desired. Overall, we have made two orthogonal projections: a projection from the distributions over graphs to the distributions over nodes, and a projection from the distribution over nodes to the product distributions $u \times \cdots \times u$. Since the product distributions forms a linear space contained in the distributions over nodes, these two projections are equivalent to a single orthogonal projection from the distributions over graphs to the product distributions over nodes. A similar reasoning holds for edges.

## E  SUBSTRUCTURE CONDITIONED GENERATION

Given a subgraph $S = (\boldsymbol{X}_S, \mathbf{E}_S)$ with $n_s$ nodes, we can condition the generation on $S$ by masking the generated node and edge feature tensor at each reverse iteration step (Lugmayr et al., 2022). As our model is permutation equivariant, it does not matter what entries are masked: we therefore choose the first $n_s$ ones. After sampling $G^{t-1}$, we update $\boldsymbol{X}$ and $\mathbf{E}$ using

$$\boldsymbol{X}^{t-1} = \boldsymbol{M}_X \odot \boldsymbol{X}_s + (1 - \boldsymbol{M}_X) \odot \boldsymbol{X}^{t-1} \quad \text{and} \quad \mathbf{E}^{t-1} = \mathbf{M}_E \odot \boldsymbol{E}_s + (1 - \mathbf{M}_E) \odot \mathbf{E}^{t-1},$$

where $\boldsymbol{M}_X \in \mathbb{R}^{n \times a}$ and $\boldsymbol{M}_E \in \mathbb{R}^{n \times n \times b}$ are masks indicating the $n_s$ first nodes. In Figure 8, we showcase an example for molecule generation: we follow the setting proposed by (Maziarz et al., 2022) and generate molecules starting from a particular motif called 1,4-Dihydroquinoline[2].

## F  EXPERIMENTAL DETAILS AND ADDITIONAL RESULTS

### F.1  ABSTRACT GRAPH GENERATION

**Metrics**  The reported metrics compare the discrepancy between the distribution of some metrics on a test set and the distribution of the same metrics on a generated graph. The metrics measured are degree distributions, clustering coefficients, and orbit counts (it measures the distribution of all substructures of size 4). We do not report raw numbers but ratios computed as follows:

$$r = \mathrm{MMD}(generated, test)^2 \, / \, \mathrm{MMD}(training, test)^2$$

The denominator $\mathrm{MMD}(training, test)^2$ is taken from the results table of SPECTRE (Martinkus et al., 2022). Note that what the authors report as MMD is actually MMD squared.

**Community-20**  In Table 5, we also provide results for the smaller Community-20 dataset which contains 200 graphs drawn from a stochastic block model with two communities. We observe that DiGress performs very well on this small dataset.

### F.2  QM9

**Metrics**  Because it is the metric reported in most papers, the validity metric we report is computed by building a molecule with RdKit and trying to obtain a valid SMILES string out of it. As explained by Jo et al. (2022), this method is not perfect because QM9 contains some charged molecules which would be considered as invalid by this method. They thus compute validity using a more relaxed definition that allows for some partial charges, which gives them a small advantage.

---

[2]https://pubchem.ncbi.nlm.nih.gov/compound/1_4-Dihydroquinoline

Table 5: Results on the small Community-20 dataset.

| | Degree↓ | Clustering↓ | Orbit↓ | Ratio↓ |
|---|---|---|---|---|
| GraphRNN | 4.0 | 1.7 | 4.0 | 3.2 |
| GRAN | 3.0 | 1.6 | 1.0 | 1.9 |
| GG-GAN | 4.0 | 3.1 | 8.0 | 5.5 |
| SPECTRE | 0.5 | 2.7 | 2.0 | 1.7 |
| DiGress | 1.0 | 0.9 | 1.0 | 1.0 |

Table 6: Ablation study on QM9 with explicit hydrogens. Marginal transitions improve over uniform transitions, and spectral and structural features further boost performance.

| Model | Valid↑ | Unique↑ | Atom stable↑ | Mol stable↑ |
|---|---|---|---|---|
| Dataset | 97.8 | 100 | 98.5 | 87.0 |
| ConGress | $86.7_{\pm 1.8}$ | $\mathbf{98.4}_{\pm 0.1}$ | $97.2_{\pm 0.2}$ | $69.5_{\pm 1.6}$ |
| DiGress (uniform) | $89.8_{\pm 1.2}$ | $97.8_{\pm 0.2}$ | $97.3_{\pm 0.1}$ | $70.5_{\pm 2.1}$ |
| DiGress (marginal) | $92.3_{\pm 2.5}$ | $97.9_{\pm 0.2}$ | $\mathbf{97.3}_{\pm 0.8}$ | $66.8_{\pm 11.8}$ |
| DiGress (marg. + features) | $\mathbf{95.4}_{\pm 1.1}$ | $97.6_{\pm 0.4}$ | $\mathbf{98.1}_{\pm 0.3}$ | $\mathbf{79.8}_{\pm 5.6}$ |

**Ablation study**    We perform an ablation study in order to highlight the role of marginal transitions and auxiliary features. In this setting, we also measure atom stability and molecule stability as defined in (Hoogeboom et al., 2022). Results are presented in Figure 6.

**Novelty**    We follow Vignac & Frossard (2021) and don't report novelty for QM9 in the main table. The reason is that since QM9 is an exhaustive enumeration of the small molecules that satisfy a given set of constrains, generating molecules outside this set is not necessarily a good sign that the network has correctly captured the data distribution. For the interested reader, DiGress achieves on average a novelty of $33.4\%$ on QM9 with implicit hydrogens, while ConGress obtains $40.0\%$.

## F.3    MOSES AND GUACAMOL

**Datasets**    For both MOSES and GuacaMol, we convert the generated graphs to SMILES using the code of Jo et al. (2022) that allows for some partial charges.

We note that GuacaMol contains complex molecules that are difficult to process, for example because they contain formal charges or fused rings. As a result, mapping the train smiles to a graph and then back to a train SMILES does not work for around $20\%$ of the molecules. Even if our model is able to correctly model these graphs and generate graphs that are similar, these graphs cannot be mapped to SMILES strings to be evaluated by GuacaMol. More efficient tools for processing complex molecules as graphs are therefore needed to truly achieve good performance on this dataset.

**Metrics**    Since MOSES and Guacamol are benchmarking tools, they come with their own set of metrics that we use to report the results. We briefly describe this metrics: Validity measures the proportion of molecules that pass basic valency checks. Uniqueness measures the proportion of molecules that have different SMILES strings (which implies that they are non-isomorphic). Novelty measures the proportion of generated molecules that are not in the training set. The filter score measures the proportion of molecules that pass the same filters that were used to build the test set. The Frechet ChemNetDistance (FCD) measures the similarity between molecules in the training set and in the test set using the embeddings learned by a neural network. SNN is the similarity to a nearest neighbor, as measured by Tanimoto distance. Scaffold similarity compares the frequencies of Bemis-Murcko scaffolds. The KL divergence compares the distribution of various physicochemical descriptors.

**Likelihood**    Since other methods did not report likelihood for GuacaMol and MOSES, we did not include our NLL results in the table neither. We obtain a test NLL of 129.7 on QM9 with explicit hydrogens, 205.2 on MOSES (on the separate scaffold test set) and 308.1 on GuacaMol.

Table 7: Proportion of valid and unique molecules obtained when sampling larger molecules than the maximal size in the training set. Interestingly, DiGress performs very well on GuacaMol and poorly on MOSES. We hypothesize that this is due to GuacaMol being a more diverse dataset, which forces the network to learn to generate good molecules of all sizes.

|  | Dataset statistics | | | Valid and unique (%) | | |
|---|---|---|---|---|---|---|
|  | $n_{\min}$ | $n_{\text{average}}$ | $n_{\max}$ | $n_{\max} + 5$ | $n_{\max} + 10$ | $n_{\max} + 20$ |
| MOSES | 8 | 21.7 | 27 | 2.6 | 2.2 | 0.0 |
| GuacaMol | 2 | 27.8 | 88 | 87.3 | 85.6 | 80.5 |

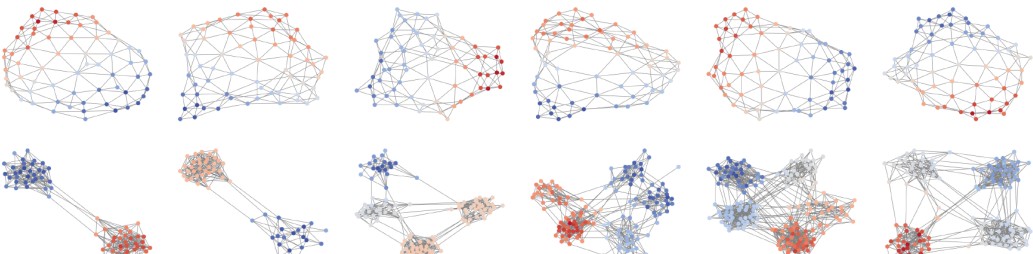

Figure 9: Non curated samples generated by DiGress trained on planar graphs (top) and graphs drawn from the stochastic block model (bottom).

**Size extrapolation** While the vast majority of molecules in QM9 have the same number of atoms, molecules in MOSES and Guacamol have varying sizes. On these datasets, we would like to know if DiGress can generate larger molecules than it has been trained on. This problem is usually called size extrapolation in the graph neural network literature.

To measure the network ability to extrapolate, we set the number of atoms to generate to $n_{\max} + k$, where $n_{\max}$ is the maximal graph size in the dataset and $k \in [5, 10, 20]$. We generate 24 batches of 256 molecules (=6144 molecules) in each setting and measure the proportion of valid and unique molecules – all these molecules are novel since they are larger than the training set.

The results are presented in Table 7. We observe an important discrepancy between the two datasets: DiGress is very capable of extrapolation on GuacaMol, but completely fails on MOSES. This can be explained by the respective statistics of the datasets: MOSES features molecules that are relatively homogeneous in size. On the contrary, GuacaMol features molecules that are much larger than the dataset average. The network is therefore trained on more diverse examples, which we conjecture is why it learns some size invariance properties. The major difference in extrapolation ability that we obtain clearly highlights the value of large and diverse datasets.

We finally note that our denoising network was not designed to be size invariant, as it for example features sum aggregation functions at each layer. Specific techniques such as SizeShiftReg (Buffelli et al., 2022) could also be used to improve the size-extrapolation ability of DiGress if needed for downstream applications.

## G  SAMPLES FROM OUR MODEL

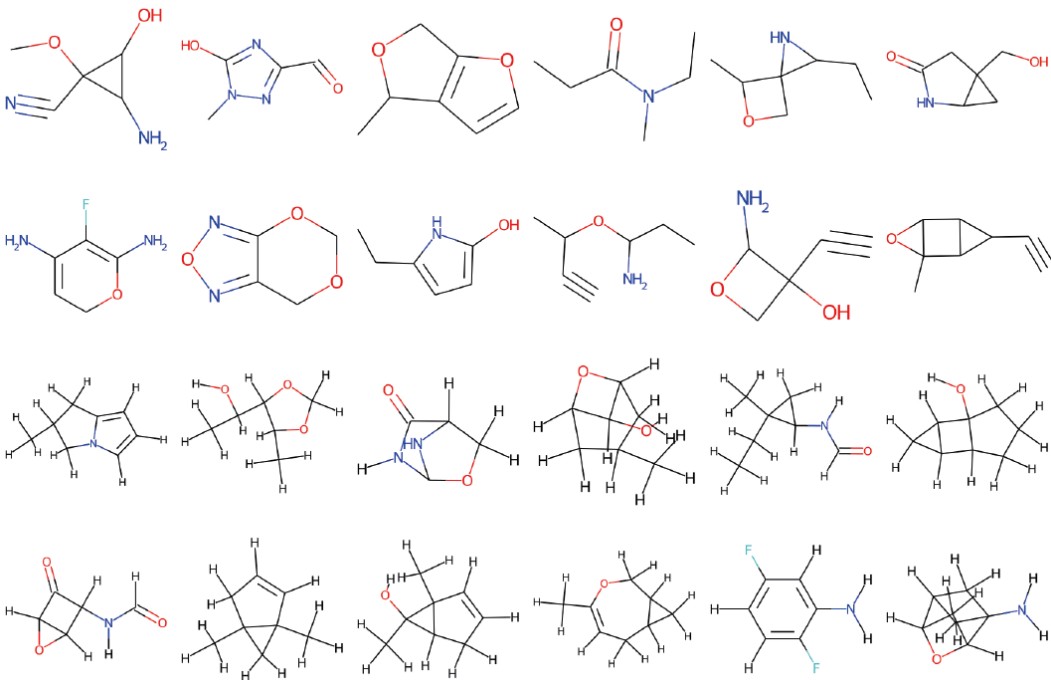

Figure 10: Non curated samples generated by DiGress, trained on QM9 with implicit hydrogens (top), and explicit hydrogens (bottom).

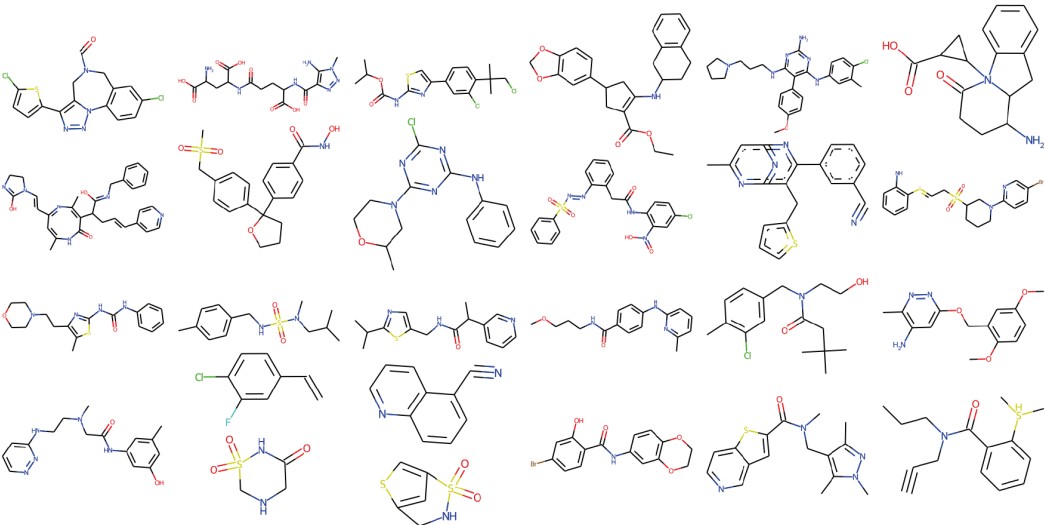

Figure 11: Non curated samples generated by Guacamol (top) and Moses (bottom). While there are some failure cases (disconnected molecules or invalid molecules), our model is the first non autoregressive method that scales to these datasets that are much more complex than the standard QM9.

