# OpenReview forum: "DiGress: Discrete Denoising diffusion for graph generation"
_ICLR.cc/2023/Conference — ICLR 2023 poster_

### Official Review · Reviewer_7EF5 · 2022-10-24

**Confidence:** 3
**Correctness:** 4
**Technical Novelty And Significance:** 3
**Empirical Novelty And Significance:** 4
**Recommendation:** 8

**Clarity, Quality, Novelty And Reproducibility:**

Clarity and quality: I was left confused about how the graph transformer network maintains a discrete graph before checking the code. Other than that, the paper is generally well-written; the literature review is thorough and complete.
Novelty: Using discrete diffusion in graph generation is technically novel. Conditioning a one-shot generation model on a subgraph is, to the best of my knowledge, new to the literature.
Reproducibility: I briefly checked the code. It appears to be well written.

**Strength And Weaknesses:**

=== Strength ===

This paper fills a known gap in the literature, between discrete diffusion [Austin 2021] and diffusion for graph data [Niu 2020]. As a result, the approach is well-motivated, and contributes greatly to the literature on deep graph generation. The model is analyzed thoroughly, especially on permutation equivariance and invariance. The ablation studies are thorough: the benefit of discrete (over continuous) diffusion, marginal (over Gaussian) noise, and conditional (over unconditional) generation are well-established.

=== Minor weaknesses ===

What is z in the algorithms?

I understand the omission of graph transformer network, but the discussion about sampling after adding noise should not be omitted. After all, it is the main reason why the diffusion model is "discrete."

I would like to see an evaluation on training on small graphs and testing on large graphs, similar to those in "Learning the Travelling Salesperson Problem Requires Rethinking Generalization".

**Summary Of The Paper:**

This paper proposes a discrete diffusion model for graph generation by combining discrete diffusion [Austin 2021] and graph transformer [Dwivedi and Bresson 2021]. Its main novelties are:
 - The diffusion model is discrete: sampling of node and edge features is performed after adding noise.
 - The denoising network is a graph transformer. It is permutation equivariant and generalizes to graphs of different sizes (sec 3.3).
 - Since "having no edge" is encoded as a categorical edge feature, choosing a uniform noise model does not preserve the sparsity of the graph. Instead, the paper proposes a noise model that preserves the marginal distribution of node and edge types during.
 - Conditional generation is made possible through regressor guidance.

**Summary Of The Review:**

The paper is well-written, the contribution is novel, and should be of interest to a broad graph generation and diffusion audience. The method proposed is well-motivated and the empirical evaluation is thorough. I recommend accept.

---

> ### Author Response · Authors · 2022-11-10
> **Answer to reviewer 7EF5 1/1**
>
> Thank you for your comments and requests for clarifications. We address your concerns below:
>
> **What is z in the algorithms?**
> Thank you for noticing that z is not defined. In section 2, z refers to noisy data. We have updated the paper accordingly.
>
> **I understand the omission of graph transformer network, but the discussion about sampling after adding noise should not be omitted. After all, it is the main reason why the diffusion model is "discrete."**
>
> We have updated the paper to better highlight that all the distributions that we consider are categorical distributions, and that sampling from the noise model means sampling classes (i.e., discrete objects) according to the computed probabilities. Thanks for pointing this out.
>
> **I would like to see an evaluation on training on small graphs and testing on large graphs, similar to those in "Learning the Travelling Salesperson Problem Requires Rethinking Generalization".**
>
> This is a very nice suggestion. We are currently running experiments on MOSES in size extrapolation settings. We will get back to you as soon as we have some results.
>
> We would be happy to further answer your concerns if some points are not clear.

---

> > ### Author Response · Authors · 2022-11-11
> > **Results on size extrapolation**
> >
> > Dear reviewer 7EF5,
> >
> > we have obtained results on size extrapolation. You can find the procedure and results below -- these results have been added to the paper as well:
> >
> > **Procedure**:
> >
> > To measure the network ability to extrapolate, we set the number of atoms to generate to $n_\text{max} + k$, where $n_\text{max}$ is the maximal graph size in the dataset and $k \in [5, 10, 20]$. We generate 24 batches of 256 molecules (=6144 molecules) in each setting and measure the proportion of valid and unique molecules -- all these molecules are novel since they are larger than the training set.
> >
> > **Results**:
> >
> > The results are presented in the table below. We observe an important discrepancy between the two datasets: DiGress is very capable of extrapolation on GuacaMol, but completely fails on MOSES. This can be explained by the respective statistics of the datasets: while MOSES features molecules that are relatively homogeneous in size, GuacaMol features molecules that are much larger than the dataset average. The network is therefore trained on more diverse examples, which we conjecture is why it learns some size invariance properties. The major difference in extrapolation ability that we obtain clearly highlights the value of large and diverse datasets.
> >
> > |          | Dataset statistics |           |       | Valid and unique (%) |            |            |
> > |----------|--------------------|-----------|-------|----------------------|------------|------------|
> > |    | n_min              | n_average | n_max | n_max + 5            | n_max + 10 | n_max + 20 |
> > | MOSES    | 8                  | 21.7      | 27    | 2.6                  | 2.2        | 0.0        |
> > | GuacaMol | 2                  | 27.8      | 88    | 87.3                 | 85.6       | 80.5       |
> >
> >
> > We note that our denoising network was not designed to be size invariant, as it for example features sum aggregation functions at each layer. Specific techniques such as SizeShiftReg [1] could also be used to improve the size-extrapolation ability of DiGress if required for downstream applications.
> >
> > [1]: Buffelli et al. 2022, SizeShiftReg: a Regularization Method for Improving Size-Generalization in Graph Neural Networks

---

### Official Review · Reviewer_br8Y · 2022-10-24

**Confidence:** 3
**Correctness:** 4
**Technical Novelty And Significance:** 3
**Empirical Novelty And Significance:** 3
**Recommendation:** 6

**Clarity, Quality, Novelty And Reproducibility:**

Clarity (High): the paper is written clearly in general.

Quality (High): The paper has extensive method and experiment sections that are generally well written.

Originality (High): while training diffusions on categorical data is not exactly new, it is interesting to choose a different forward distribution. This is also well motivated in the context of graph generation.

Reproducibility (High): code is provided.

**Strength And Weaknesses:**

Strength:
- The paper is written clearly and the idea behind non-uniform diffusion noise addition makes sense, and is empirically confirmed.
- The proposed architecture handles node-level, edge-level and graph level features and has some nice properties, such as equivariance, and exchangeability.
- The diffusion model framework allows for easy use of classifier-based / classifier-free guidance for conditional generation.

Weakness:
- The method does not deal with growing graphs like in autoregressive models.
- The sampling procedure is not entirely clear (in algorithm 3). While the distribution $p(G^{t-1} | G^t) p(y | G^{t-1})$ can be sample from Langevin dynamics, it is not sure if that is actually used -- more likely it is similar to the sampling procedure of diffusion models, which would not guarantee that $G^{t-1}$ is sampled from that distribution described above.
- Some of the experimental results seem odd. In Table 1. the V.U.N. for ConGress is 0% meaning that it generates nothing that is valid, unique and novel graphs. In Table 4, the validity of Congress is 0% and FCD is also 0, which seems quite odd because FCD is better if it is lower (that is also a typo? also KL div is pointed upwards as well).
- The performance between ConGress and DiGress differ by a lot, which seems odd as usually the discrete case converges to continuous case as number of timesteps goes to infinity.

**Summary Of The Paper:**

The paper presents DiGress, a diffusion model for graph generation. Apart from using graph-based architectures, the key idea is to start with a noise model that preserves the marginal distribution of node and edge types rather than using uniform noise. Then the denoising network is augmented with structural features. Finally, conditional generation is enabled by guidance procedures. Empirical results shows that the performance of DiGress matches the performance of autoregressive models on some molecular datasets.

**Summary Of The Review:**

The method section of the paper is written nicely, but there are some confusions surrounding the experimental sections.

---

> ### Author Response · Authors · 2022-11-10
> **Answer to reviewer br8y 1/2**
>
> Thank you for your insightful comments and requests for clarifications. We address your concern below:
>
> **The performance between ConGress and DiGress differ by a lot, which seems odd as usually the discrete case converges to continuous case as number of timesteps goes to infinity.**
>
> We would like to start by answering this comment, because this clarification might help address your other comments as well.
>
> When we say that DiGress is a discrete diffusion model, we do not mean that it operates on discrete time steps (although this is true as well). We mean that i) it manipulates objects on a discrete state-space, ii) that it samples discrete noise during training iii) at each iteration of sampling, it samples discrete graphs as well. We have updated the text to better highlight this aspect.
>
> There is therefore no condition under which Congress converges to Digress. These two models are very different in the noise that they use and the task that they solve. For example, DiGress solves node and edge classification problems, while ConGress solves node and edge regression problems.
>
> While it is true that ConGress performs very poorly on the large datasets, we believe that our implementation is correct. DiGress apart, our ConGress baseline is for example state-of-the-art on QM9. Across our experiments, we observed the general trend that the gap between DiGress and ConGress gets bigger when working on larger graphs. We believe that this is because DiGress directly operates on the space of discrete graphs, without solving a continuous surrogate problem as ConGress or GDSS do.
>
> **The method does not deal with growing graphs like in autoregressive models**
>
> This is indeed true. We consider that our method is a one-shot model because it is autoregressive in time, but not in nodes.
>
> Whether or not this is a good thing depends on the application. When the true data generation process corresponds to graphs that are growing, then autoregressive models are likely to perform very well. When it is not the case, they often do not. For example, MolRNN performs very badly on the planar graph dataset (Table 1).
>
> One general drawback of autoregressive models is that they introduce an ordering of the nodes that usually does not exist in the data. There is therefore a lot of interest in the design of methods that respect permutation equivariance (as DiGress does). We are happy to be the first one-shot method that operates at the node level, which performs as well as autoregressive models at molecule generation.
>
> Finally, we note that it is possible to grow an existing graph with our model, as is done in Appendix D.1 (with results in Figure 7).

---

> > ### Author Response · Authors · 2022-11-10
> > **Answer to reviewer br8y 2/2**
> >
> >
> > **Some of the experimental results seem odd, in particular i) Congress in Table 1, and ii) the FCD and KL div for Guacamol.**
> >
> > As explained above, we believe that the results of ConGress are correct. The graphs of Table 1 are large graphs (larger than the molecules in the other datasets), which is why many models achieve 0% validity. When inspecting the generated graphs visually, we observe that the graphs generated by ConGress are far from planar graphs.
> > The only exception is SPECTRE, which performs well on these datasets. The reason is that it manipulates graph spectral features, which are very good at capturing long range dependencies in the graphs. SPECTRE is therefore well suited to large graphs, but it is not really able to produce structures that are locally good. This is visible on the QM9 dataset where it performs quite poorly.
> >
> > There is no typo in the results of GuacaMol. GuacaMol is a benchmarking tool that returns scores (higher is better) between 0 and 1 for all metrics. This is true for FCD and KL divergence as well. We have updated the paper to make this clear.
> >
> >
> > **The sampling procedure of algorithm 3 is not entirely clear. Is it guaranteed to sample from the distribution described above?**
> >
> > We are not sure to understand this question exactly. Could you please have the kindness to clarify your concern?
> >
> > We however want to highlight the following point: since the space that we consider is discrete, the distributions that we manipulate are categorical distributions. We can compute the probabilities for each class and sample exactly from this distribution.
> >
> >
> >
> > We hope to have convinced you that our experiments are performed on difficult tasks on which most previous models do not work, and that both the diffusion architecture and the discrete state-space are required to make things work on these tasks. On the simple QM9 dataset that contains small graphs, our Gaussian-based baseline works well, but it does not scale as nicely as DiGress to larger graphs and more difficult tasks. DiGress does not solve a continuous surrogate problem, it can adapt to the sparsity level of the data and can leverage additional features which cannot be computed with the other methods. Together, these properties result in faster training and a better ability to solve complex problems.
> >
> > We would be happy to further answer your concerns if some points are not clear.

---

### Official Review · Reviewer_SeWu · 2022-10-28

**Confidence:** 4
**Correctness:** 4
**Technical Novelty And Significance:** 3
**Empirical Novelty And Significance:** 3
**Recommendation:** 6

**Clarity, Quality, Novelty And Reproducibility:**

Overall, the paper reads well and tackles a relevant problem (one-shot molecule generation). Nonetheless, the technical novelty is limited as the work follows the framework by Austin et al., (2021), with improvements mainly coming from adopting marginal probabilities as prior and structural features. Regarding reproducibility, the authors have made their code available, and the paper provides sufficient details about the method and its training methodology.

A few questions/comments:
- It requires some effort to understand some parts of the formulation. For instance, the idea of predicting the clean graph at each step $t$ isn't adequately motivated and doesn't fit into the formulation introduced in Section 2. Also, the paper uses $u$ and $v$ to refer to node and edge distributions, which I found unusual.
- ZINC represents one of the main benchmarks to evaluate molecule generation. Any specific reason for not considering it?
- What are the numbers for VUN (valid, unique & novel) for the experiments in Tables 3 and 4?
- Tables 3 and 4 miss some baselines --- e.g., GraphAF (https://arxiv.org/pdf/2001.09382.pdf).
- While the paper shows the idea is promising, the results are not strong. For instance, in Table 3, JT-VAE (a contribution from 2018) outperforms DiGress and ConGress. Also, LSTM on SMILES representation (used in the REINVENT tool, for instance) beats the proposal. - - Thus, the paper doesn't seem to advance the current art.
- Could the same additional features be used to boost the performance of the baselines?
- Experiments on conditional generation are weak. What are the current SOTA methods for these tasks? How do the proposed methods perform on different properties?
- Typo: $G^t$ instead of $G_t$ in Algorithm 1.

**Strength And Weaknesses:**

Strengths
- The paper also introduces a continuous version (ConGress) and uses it for ablation purposes, showing that the discrete version produces better results.
- The proposed method yields good results for general graph generation, outperforming SPECTRE (ICML, 2022).

Weaknesses
- The proposed method does not advance the current art --- widely used molecule generation methods (e.g., JT-VAE) outperform the proposed approaches.
- Experiments on conditional generation are somewhat weak, with no extensive comparison with other baselines.

**Summary Of The Paper:**

The paper introduces DiGress, a discrete diffusion-based generative model for graphs. The noise model sequentially corrupts the graph structure by modifying the categories of edges and nodes. The denoising model is a Transformer that leverages additional features. Experiments on graph and (conditional) molecule generation demonstrate the efficacy of DiGress.

**Summary Of The Review:**

Although I have concerns regarding clarity and performance gains over current art, this is an interesting contribution to one-shot molecular generation. Thus, I am inclined to accept the paper.

---

> ### Author Response · Authors · 2022-11-10
> **Answer to reviewer SeWu 1/2**
>
> We would like to thank the reviewer for the useful comments and suggestions. We will first address the reviewer concerns about the algorithm description and motivation, before discussing experimental results.
>
> **It requires some effort to understand some parts of the formulation, such as the idea of predicting the clean graph at each step.**
>
> We have modified Section 2 to better reflect this motivation. The reason behind not training the network to predict $G^{t-1}$ from $G^{t}$ is that $G^{t-1}$ is a very noisy object: it depends both on the sampling of the clean graph $G$ in the training set, and on the noise trajectory $q(G^{t-1} | G)$ that is sampled. By training the network to predict directly the clean graph (as done in DiGress), we remove the noise due to the trajectory sampling. The network is therefore trained on a less noisy target, which seems to be key to its efficiency. This is a major difference between early diffusion models [1] and modern diffusion models [2] and it seems to explain the major boost in performance obtained with diffusion models in the two past years.
>
> While virtually all modern diffusion models either predict the clean input or the noise that has been added to the clean input, we found no explanation in the literature on why this is important. We hope that our explanation will clarify this point for future diffusion models as well.
>
> [1]: Sohl-Dickstein et al. 2015, Deep Unsupervised Learning using Nonequilibrium Thermodynamics
> [2]: Ho et al. 2020, Denoising Diffusion Probabilistic Models
>
> **The paper uses and refers to node and edge distributions, which I found unusual.**
>
> Our work indeed uses what we call the “marginal distribution of node and edge types”, i.e. the frequency of each node and edge type in the training set. If you would like to suggest a more appropriate terminology, we would be happy to change the text accordingly.
>
> **Could the same additional features be used to boost the performance of the baselines?**
>
> These additional features can only be computed when i) the objects in the latent space can be interpreted as noisy graphs, and ii) these graphs are sparse, so that cycle computations or spectral features are meaningful. Among the models that we report, only DiGress satisfies these two properties. Our additional features can therefore not be used in a straightforward manner with the other methods.
>
> **ZINC represents one of the main benchmarks to evaluate molecule generation. Any specific reason for not considering it?**
>
> We indeed do not use the ZINC 250k database. Instead, we used MOSES, which is another subset of the ZINC database. MOSES was gathered by computational chemists and contains many interesting ideas that we would like to promote in the machine learning community. For example:
>   - It only contains molecules that have successfully passed a number of filters (for example, they do not contain substructures that are known to be toxic). The same filters can be applied on the generated molecules in order to check that the model has successfully learned these rules. This is the Filters metric that we report.
>   - The test set of MOSES was designed to contain molecular scaffolds (i.e, ring structures) that were not observed in the training set. One of the metrics (Scaf in Table 3) measures the ability to generate scaffolds that are realistic (since they are present in the test set) but that were not observed in the training set. This prevents models that only marginally edit molecules in the training set to score high on all metrics.
>   - MOSES provides quite a lot of metrics that together give a good overview of the performance of the generative model.
>
> Furthermore, we also consider the GuacaMol dataset which is based on CHEMBL rather than ZINC. CHEMBL tends to contain larger and more complex molecules (that have been synthesized but are not necessarily commercially available), so we believe that GuacaMol will also become an important tool for benchmarking models in the future.
>
> If you consider that results on ZINC 250k would really be beneficial for the paper, we should be able to run additional experiments in the coming week. We expect results to be similar to the other results on molecules, with a validity in the range of 85-100% and a high rate of unique and novel molecules.
>
> **Tables 3 and 4 miss some baselines --- e.g., GraphAF**
>
> We made the deliberate choice to not include models that perform validity correction in the tables. While validity correction is useful for downstream applications, we believe that it obfuscates the performance of the learned generative model, and that it should not be used to compare models.
>
> Regarding GraphAF specifically, the authors report a validity without correction of 67% on QM9 and 71% on MOSES, which are both below DiGress (99% on QM9, 86% on MOSES).
> The other metrics are reported after validity correction, which is why we did not include GraphAF in our tables.

---

> > ### Author Response · Authors · 2022-11-10
> > **Answer to reviewer SeWu 2/2**
> >
> > **While the paper shows the idea is promising, the results are not strong. For instance, in Table 3, JT-VAE outperforms DiGress and ConGress. Also, LSTM on SMILES representation beats the proposal. - - Thus, the paper doesn't seem to advance the current art.**
> >
> >  DiGress is a graph generation model that operates at the node level, and we believe that it should primarily be compared to other models that belong to the same class. It is perfectly possible to envision discrete denoising diffusion models that leverage the junction tree or molecular fragments, but it seems natural to define the vanilla model first. Our experimental results clearly show that DiGress outperforms other one-shot node-level models by a large margin, which is why we are convinced that it advances the state of the art.
> >
> > Furthermore, DiGress is a general graph generation model that is not only targeted to molecules. Consider for example the planar graphs of Table 1:
> >   - For these graphs the junction tree is made of a single node, so that JTVAE would probably not perform well.
> >   - Autoregressive models perform very poorly on these graphs, as can be seen with GraphRNN. The reason is that the inductive bias of recurrent graph networks (namely, that the graph was built by recursively attaching nodes) is not at all suited to this dataset.
> >   - DiGress clearly outperforms other methods, including the most recent ones (SPECTRE or GG-GAN).
> >
> > Finally, regarding molecule generation, it is true that to this day the most efficient representation seems to be SMILES strings rather than graphs. The main reason is that SMILES incorporate a lot of information about chemistry, so that the model does not have to learn the rules of chemistry by itself. There are however many tasks that can be solved by graph-based models but not by SMILES, such as conformer generation or docking prediction. This is why so much effort is devoted to building new graph-based methods.
> > While DiGress currently does not address these tasks, it could be modified to include 3d information and conditionally generate molecules that bind to some proteins of choice.
> >
> > **What are the numbers for VUN (valid, unique & novel) for the experiments in Tables 3 and 4?**
> >
> > These numbers are computed in such a way that the fraction of molecules that are valid, unique and novel is simply the product of the three reported metrics. For DiGress, it is 81.4% on MOSES and 85.1% on GuacaMol, while for ConGress it is 80.3 on MOSES and only 0.1 on GuacaMol. We recall that the other methods that we compare to on these datasets are targeted to molecules, and that such methods generate valid molecules by design, which makes the comparison of VUN alone relatively unfair.
> >
> > **Experiments on conditional generation are weak. What are the current SOTA methods for these tasks? How do the proposed methods perform on different properties?**
> >
> > We see the results on conditional generation as a proof-of-concept that conditional generation is possible with discrete diffusion models. Our guidance procedure does significantly improve over the unconditional model, but there is clear room for improvement as the reported mean absolute errors are far from zero.
> >
> > Unfortunately, there is no well-established baseline for conditional generation. Several papers focus on molecule optimization and adopt a latent state optimization approach or Monte-Carlo sampling to maximize a target property such as solubility, drug-likeness or molecular weight [1, 2, 3]. The main difficulty with such procedures is to maximize the property of interest without straying away too much from the train data distribution.
> >
> >
> > The two works that are closest to our proposition are [4, 5]. These works use reinforcement learning to generate molecules with a specified molecular weight or solubility. Their evaluation protocol is based on single-property conditioning, and the values for these properties are arbitrarily picked. If we were to apply multiple-property conditioning with arbitrarily picked values, we would have no guarantee that there exist molecules that satisfy all the properties simultaneously.
> >
> > On the contrary, we choose the conditioning values using molecules in the test set. We therefore have the guarantee that molecules with the target properties exist. We thus hope that our evaluation protocol will be used in future work, and that better tools will be proposed that outperform our guidance conditioning.
> >
> > We would be happy to further address your concerns if some points are not clear.
> >
> > [1] Richards and Groener 2022,  Conditional beta-VAE for de novo molecular generation
> > [2] Lee et al. 2022,  Exploring chemical space with score-based out-of-distribution generation
> > [3] Wang et al. 2022, Relation: a deep generative model for structure-based de novo design
> > [4] Mahmood et al. 2021, Masked graph modeling for molecule generation
> > [5] Kwon et al. 2019, Efficient learning of non-autoregressive graph variational autoencoders for molecular graph generation

---

### Author Response · Authors · 2022-11-10
**Answer to all reviewers 1/1**

We would like to thank the reviewers for their comments and suggestions. We have updated the paper based on your reviews. The main changes are listed below:

  - We have reformulated Section 2 to better explain why training the neural network to predict the clean graph is more efficient than training it to predict G^{t-1}.
  - In Section 3, we have made it more clear that we are operating on a discrete space – that when we sample noisy data, this data is a discrete object, and that when we sample G^{t-1} from G^t, this is a discrete object as well.
  - A typo in the equations for the reverse process of Section 3 was fixed.
  - Section 3.2 was reformulated to better motivate the choice of a graph transformer network.
  - We added a reference to a concurrent (but less comprehensive) discrete diffusion model that was made public after the paper submission.
  - We highlighted that the GuacaMol number are scores, so that higher is better
  - We ran a larger model on GuacaMol which yields better results.

We will also add the size extrapolation results in Appendix when they are available.

We believe that this new version makes it more clear that DiGress operates on a discrete state-space and that the distributions that are considered are categorical distributions over this discrete space. We hope to have addressed all your concerns. If it is not the case, we will be happy to further answer your comments.

---

### Public Comment · ~Divin_Yan1 · 2022-11-15
**Wondering what type of GPU you are using for this work.**

Wondering what type of GPU you are using for this work,thanks!

---

> ### Author Response · Authors · 2022-11-15
> **GPUs used in DiGress**
>
> Hello Liang,
>
> we used Tesla V-100s. The experiments on GuacaMol and MOSES were launched on 3 gpus, but for the other experiments we used a single gpu.
>
> Best,
>
> the authors.

---

### Decision · Program_Chairs · 2023-01-20

**Decision:**

Accept: poster

**Justification For Why Not Higher Score:**

The paper can be viewed as a relatively straightforward combination of two existing techniques: the discrete diffusion of Austin et al. (2021) and graph transformer of Dwivedi and Bresson (2021). A major theoretical or methodological contribution will help improve the score.

**Justification For Why Not Lower Score:**

As well summarized by Reviewer 7EF5: "This paper fills a known gap in the literature, between discrete diffusion [Austin 2021] and diffusion for graph data [Niu 2020]. As a result, the approach is well-motivated, and contributes greatly to the literature on deep graph generation. The model is analyzed thoroughly, especially on permutation equivariance and invariance. The ablation studies are thorough: the benefit of discrete (over continuous) diffusion, marginal (over Gaussian) noise, and conditional (over unconditional) generation are well-established."

**Metareview: Summary, Strengths And Weaknesses:**

The paper introduces a diffusion model for graph generation, referred to as DiGress. Building on top of the discrete diffusion framework of Austin et al. (2021), DiGress sequentially corrupts the graph by modifying the categories of edges and nodes. To reconstruct the graph, it borrows the graph transformer [Dwivedi and Bresson 2021] in the denoising reverse diffusion process. Different from previous discrete diffusion that starts the reverse from a uniform noise, DiGress starts with a noise model that preserves the marginal distribution of node and edge types. In addition, DiGress enables conditional generation with regressor guidance. The efficacy of DiGress is demonstrated with experiments on graph and (conditional) molecule generation, with better performance than autoregressive models on some molecular datasets. All reviewers are in favor of accepting the paper.

**Note From Pc:**

if the above contains the word "oral" or "spotlight" please see: "oral" presentation means -> notable-top-5% and "spotlight" means -> notable-top-25%. As stated in our emails, we are disassociating presentation type from AC recommendations